# DeAltHDR: Learning HDR Video Reconstruction from Degraded Alternating Exposure Sequences

**Shuohao Zhang**[1], **Zhilu Zhang**[1]*, **Rongjian Xu**[2], **Xiaohe Wu**[1], **Wangmeng Zuo**[1]

[1]Harbin Institute of Technology, Harbin, China
[2]City University of Hong Kong, Hong Kong

`awesomezshao@gmail.com, cszlzhang@outlook.com, ronjon.xu@gmail.com,`
`csxhwu@gmail.com, wmzuo@hit.edu.cn`

## Abstract

High dynamic range (HDR) video can be reconstructed from low dynamic range (LDR) sequences with alternating exposures. However, most existing methods overlook the degradations (e.g., noise and blur) in LDR frames, focusing only on the brightness and position differences between them. To address this gap, we propose DeAltHDR, a novel framework for high-quality HDR video reconstruction from degraded sequences. Our framework addresses two key challenges. First, noisy and blurry content complicate inter-frame alignment. To tackle this, we propose a flow-guided masked attention mechanism that leverages optical flow for a dynamic sparse cross-attention computation, achieving superior performance while maintaining efficiency. Notably, its controllable attention ratio allows for adaptive inference costs. Second, the lack of real-world paired data hinders practical deployment. We overcome this with a two-stage training paradigm: the model is first pre-trained on our newly introduced synthetic paired dataset and subsequently fine-tuned on unlabeled real-world videos via a proposed self-supervised method. Experiments show our method outperforms state-of-the-art ones. Code and data will be available at `https://zhang-shuohao.github.io/DeAltHDR/`.

## 1 Introduction

High dynamic range (HDR) imaging (Fairchild, 2007; Grosch et al., 2006; Yan et al., 2019), which is renowned for its ability to preserve details across an extensive luminance range (from deep shadows to bright highlights), offers a more immersive and realistic visual experience. This has catalyzed substantial demand for HDR content across diverse domains, including film production and mobile photography. While specialized hardware (Tocci et al., 2011; Kronander et al., 2013; Choi et al., 2017; Nayar & Mitsunaga, 2000) can capture multiple exposures simultaneously to generate HDR assets, these systems are often hindered by high costs and limited portability. Consequently, computational methods that reconstruct HDR content from asynchronous multi-exposure low dynamic range (LDR) sequences have emerged as a more practical and cost-effective alternative. A prominent task is HDR video reconstruction (Kang et al., 2003; Kalantari et al., 2013; Chen et al., 2021) from LDR frames captured with alternating short and long exposures.

However, a critical limitation of most existing HDR video reconstruction methods (Chen et al., 2021; Chung & Cho, 2023b; Cui et al., 2024) is their underlying assumption of noise-free and blur-free input frames. Consequently, their designs focus primarily on compensating for inter-frame brightness variations and spatial misalignment to mitigate ghosting artifacts, while overlooking crucial degradations. This idealized assumption rarely holds true in practice. The alternating exposure strategy can inherently introduce artifacts: short-exposure frames are often corrupted by significant noise, particularly in low-light conditions, while long-exposure frames are susceptible to motion blur from

---

*Corresponding Author.

camera shake or object movement. This gap between assumption and reality severely hinders the applicability of existing methods in real-world scenarios.

Although a recent work, BracketIRE (Zhang et al., 2025), does consider such degradations, it was specifically engineered for HDR image reconstruction. As a result, it yields suboptimal performance when directly applied to videos. In this work, we propose a novel framework to reconstruct high-quality HDR videos from degraded alternating exposure frames, named DeAltHDR. The framework addresses two key challenges. **Firstly**, inter-frame alignment is a critical but non-trivial issue due to complex object motions and occlusions. Moreover, the noise and blur degradations further increase the difficulty of alignment. In this case, the commonly used optical flow (Xu et al., 2024; Cui et al., 2024; Kong et al., 2024) and deformable convolution (Chan et al., 2022) alignment exhibit limited performance. Attention-based methods (Chung & Cho, 2023a; Tel et al., 2023a) can be effective, but their high computational complexity and large time demands present a substantial burden. To address these issues, we propose a novel Flow-Guided Masked Attention (FGMA) alignment mechanism, which integrates optical flow and attention methods flexibly and elegantly. Specifically, it first calculates a binary mask to identify 'unreliable' regions where the flow-based alignment is likely to be inaccurate. Subsequently, a cross-attention operation is applied only within these masked regions, while the rest rely on the efficient flow-based warping. This sparse and targeted application of attention achieves a superior balance between performance and computational cost. Crucially, the attention ratio can be dynamically adjustable during inference, enabling the model's computational footprint to be tailored to diverse computational budgets, as shown in Fig. 1.

**Secondly**, the scarcity of paired real-world training data presents a critical bottleneck for practical deployment. Models trained solely on synthetic data inevitably suffer from significant performance degradation when applied to real-world scenarios. To bridge this gap, we adopt the two-stage training paradigm from BracketIRE (Zhang et al., 2025): pre-training the model on synthetic paired data, followed by self-supervised fine-tuning on unlabeled real-world data. To facilitate this strategy, we propose two new datasets. For the pre-training stage, we construct a synthetic dataset by applying noise and motion blur to high-quality 4K HDR videos captured with a DJI Pocket 3. For the fine-tuning stage, we collect alternating exposure sequences with real degradations using an iPhone 16 ProMax. Nevertheless, we observe that directly applying the self-supervised fine-tuning method from BracketIRE (originally designed for HDR image reconstruction) is insufficient for the videos. It struggles to adapt to the diverse types and mag-

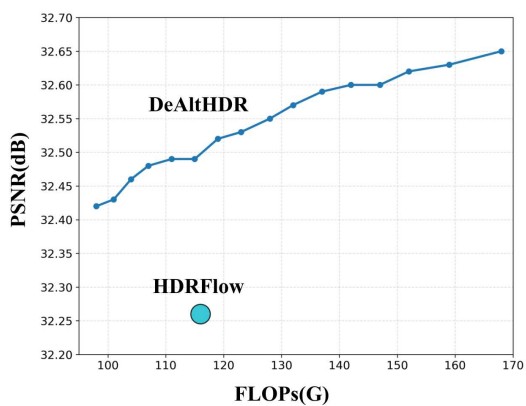

Figure 1: Comparison with HDRFlow (Xu et al., 2024), which is a representative state-of-the-art method that balances performance and efficiency. Our DeAltHDR outperforms it while the inference cost can be adjusted.

nitudes of motion present in HDR video, yielding only marginal performance gains. We therefore propose a novel motion-enhanced self-supervised adaptation method, specifically engineered to handle complex temporal dynamics. Comprehensive experiments on both our synthetic and real-world datasets validate that our proposed method significantly outperforms existing state-of-the-art ones. Our contributions can be briefly summarized as follows:

- We take the noise and blur degradations in alternating exposure frames into account, proposing a novel framework to reconstruct high-quality HDR videos from them.
- We propose a flow-guided masked attention for efficient inter-frame alignment, where the attention ratio can be dynamically adjustable for adaptive inference cost.
- We introduce a motion-enhanced self-supervised fine-tuning approach to improve the reconstruction quality on real-world videos.
- We construct synthetic and real-world datasets with rich scenes. Experiments on them show our method outperforms the state-of-the-art ones.

## 2 RELATED WORK

### 2.1 HDR IMAGE RECONSTRUCTION

HDR image reconstruction aims to render one HDR image from multiple LDR images with different exposures. Optical flow methods (Yue et al., 2023; Zimmer et al., 2011) and patch-based methods (Hu et al., 2013; Sen et al., 2012) are methods which have been proposed for aligning LDR images. However, they fail to reconstruct ghost-free HDR images with motions. With the advancement of deep learning, numerous methods (Wu et al., 2018; Liu et al., 2023; Yan et al., 2019; Zhang et al.) have been proposed for HDR image reconstruction. AHDRNet (Yan et al., 2019) uses spatial attention to guide the HDR image reconstruction to avoid the artifacts generated by optical-flow estimation error. SCTNet (Tel et al., 2023a) proposes a network with spatial and channel attentions, which aim to address the intra-image correlation for dynamic motion and the inter-image intertwining for semantic consistency across frames, respectively. SAFNet (Kong et al., 2024) focuses on finding valuable regions while estimating their easily detectable and meaningful motion for efficiency. It also devises a new window partition cropping method for training to facilitate learning on samples with large motion. Although these methods can achieve good results on HDR image reconstruction with clean LDR images, they overlook degradations in LDR images, which are common in real-world scenarios. BracketIRE (Zhang et al., 2025) takes noise and blur into account and reconstructs HDR images from degraded LDR images.

### 2.2 HDR VIDEO RECONSTRUCTION

HDR videos can be captured by specialized hardware, including scan-line or pixel exposure (Choi et al., 2017; Nayar & Mitsunaga, 2000), beam splitter (Tocci et al., 2011; Kronander et al., 2013; McGuire et al., 2007), modulo or gradient camera (Zhao et al., 2015; Tumblin et al., 2005). However, these systems are often hindered by high costs and limited portability. Therefore, computational methods that reconstruct HDR content from asynchronous multi-exposure LDR sequences have emerged as a more practical and cost-effective alternative. Kalantari *et al.* (Kalantari et al., 2013) utilizes optical flow and patch-based optimization algorithm to synthesize missing exposures for each frame. Chen *et al.* (Chen et al., 2021) introduces a coarse-to-fine deep learning framework for HDR video reconstruction consisting of coarse alignment by optical flow and more sophisticated alignment by deformable convolution. LAN-HDR (Chung & Cho, 2023a) proposes a luminance-based alignment network consisting of an alignment module and a hallucination module. Instead of optical flow, it utilizes sparse attention to align frames by evaluating luminance and color information. NECHDR (Cui et al., 2024) proposes a framework for HDR video reconstruction by reconstructing the LDR frames of absent exposures from interpolating neighbor LDR frames in the time dimension. HDRFlow (Xu et al., 2024) proposes an efficient flow estimator for real-time HDR video reconstruction which employs an HDR domain alignment loss for accurate alignment in saturated and dark regions. However, these methods mostly overlook noise and blur degradations, which are common in real-world scenarios where images are susceptible to such degradations.

## 3 METHODOLOGY

### 3.1 PROBLEM DEFINITION AND FORMULATION

In the HDR video reconstruction task, the input LDR video generally consists of LDR frames $\{\mathbf{L}_t\}_{t=1}^N$ captured under different exposures $\Delta e_t$. We aim to reconstruct a high-quality video, consisting of HDR frames $\{\mathbf{H}_t\}_{t=1}^N$. Following previous works (Xu et al., 2024; Shu et al., 2024), we configure the input frame sequence to the network with a three-stop exposure difference. Specifically, in this paper we introduce our algorithm for handling videos captured with alternating exposures and the exposure is $\{$EV-2,EV+1,EV-2, $\dots\}$. Let $\{\mathbf{L}_1, \mathbf{L}_3, \dots, \mathbf{L}_{2m-1}\}$ be short-exposure frames and $\{\mathbf{L}_2, \mathbf{L}_4, \dots, \mathbf{L}_{2m}\}$ be long-exposure frames, where $m \in \{1, 2, \dots, N/2\}$ and $N$ is an even number. Then, following previous methods focusing on multi-exposure HDR reconstruction, we normalize input luminance to ensure consistency. Firstly, we apply a simple inverse gamma correction to obtain linearized input RGB images $\{\hat{\mathbf{L}}_t\}_{t=1}^N$. Then we normalize all long-exposure frames $\{\hat{\mathbf{L}}_{2i}\}_{i=1}^{N/2}$ to $\frac{\{\hat{\mathbf{L}}_{2i}\}_{i=1}^{N/2}}{\Delta e_{2i}/\Delta e_{2i-1}}$. Therefore, we adjust the brightness of all long exposures to match

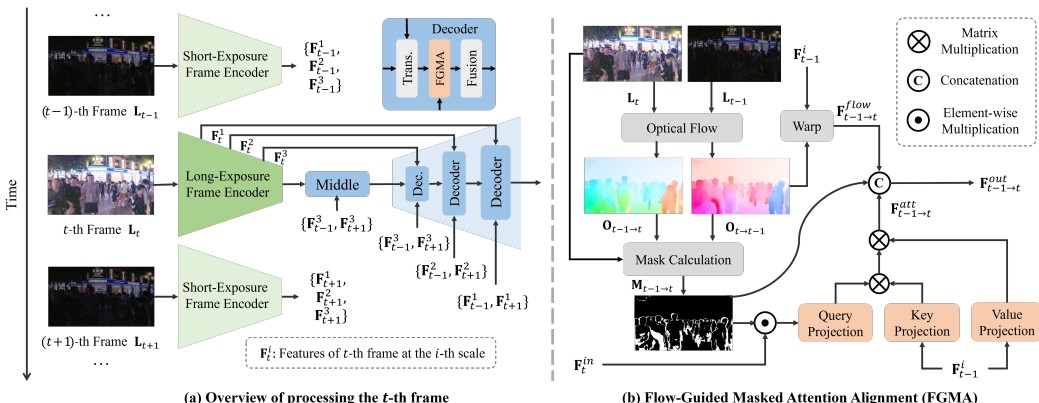

Figure 2: Overview of our framework. Figure (a) illustrates the processing of the t-th frame in our DeAltHDR, where DeAltHDR uses the other 2 neighboring frames for assistance. Taking the alignment from $t-1$-th frame to $t$-th frame as an example, figure (b) shows how Flow-Guided Mask Attention Alignment (FGMA) works.

the short exposures. For short-exposure frames, we keep them unchanged. We define the sequence formed by these new linear frames as $\{\hat{\mathbf{L}}'_t\}_{t=1}^N$. The input of the network is the concatenation of $\{\hat{\mathbf{L}}'_t\}_{t=1}^N$ and its gamma-transformed images,*i.e.*,

$$\{\mathbf{L}_t^c\}_{t=1}^N = \{\hat{\mathbf{L}}'_t, (\hat{\mathbf{L}}'_t)^\gamma\}_{t=1}^N, \tag{1}$$

where $\gamma$ represents the gamma correction parameter and is generally set to 1/2.2. Finally, we feed these concatenated images into model $\mathcal{B}$ with parameters $\Theta_\mathcal{B}$, *i.e.*,

$$\{\hat{\mathbf{H}}_\mathbf{t}\}_{t=1}^N = \mathcal{B}(\{\mathbf{L}_t^c\}_{t=1}^N; \Theta_\mathcal{B}), \tag{2}$$

where $\{\hat{\mathbf{H}}_\mathbf{t}\}_{t=1}^N$ is the generated video sequences. The key to HDR video reconstruction lies in constructing model $\mathcal{B}$ and optimizing its parameters $\Theta_\mathcal{B}$.

## 3.2 OVERVIEW OF NETWORK DESIGN

Our proposed DeAltHDR framework is based on a multi-scale encoder-decoder architecture, as shown in Fig. 2(a). We build it based on Turtle (Ghasemabadi et al., 2024), and replace its alignment module in the decoder block with our proposed flow-guided masked attention alignment module while keeping its frame history router as the fusion block. Moreover, we deploy two encoders with identical architecture for short-exposure and long-exposure frames, respectively. This will be helpful for feature extraction in a specific exposure domain.

Given an LDR frame $\mathbf{L}_t^c$, it is first fed to its corresponding encoder to extract multi-scale features $\{\mathbf{F}_t^i\}_{i=1,2,3}$, where $i$ indicates the scale of the encoder. In the $i$-th scale decoder, our proposed flow-guided masked attention alignment module takes the feature from the neighboring frame as well as the feature of the current frame as input. For the $t$-th frame, let us denote its current feature and the feature of the neighboring frame as $\hat{\mathbf{F}}_t^{in}$ and $\mathbf{F}_{t-1}^i$, respectively. Our module takes $\mathbf{F}_t^{in}$ and $\mathbf{F}_{t-1}^i$ as input to calculate the aligned neighboring features $\mathbf{F}_{t-1\to t}^{out}$, *i.e.*,

$$\mathbf{F}_{t-1\to t}^{out} = FGMA(\mathbf{F}_t^{in}, \mathbf{F}_{t-1}^i). \tag{3}$$

When implemented, we utilize the FGMA module to calculate and concatenate aligned features of 4 neighboring frames to assist in the reconstruction of the current frame's HDR information. Finally, in the fusion router block, a dynamic routing mechanism adaptively weights and combines motion-compensated neighboring features according to their relevance for current frame restoration.

## 3.3 FLOW-GUIDED MASKED ATTENTION ALIGNMENT

Inter-frame alignment plays a crucial role in video restoration. To achieve this goal, previous alignment approaches fall mainly into two categories: optical flow-based methods and attention-based

implicit alignment techniques. However, when dealing with LDR frames with noise and blur degradations, optical flow and deformable convolution alignment exhibit limited performance. Attention-based methods can achieve better quality, but with significantly higher computational costs. We also note that MIA-VSR (Zhou et al., 2024) proposed a sparse attention in video super-resolution based on a mask calculated from the difference between adjacent frames. However, this approach is less suitable for HDR video reconstruction due to the significant difference in exposure and degradation between neighboring LDR frames. In addition, the computational cost of these methods is fixed and cannot be adaptively adjusted to varying computational budgets. In this paper, we propose a novel Flow-Guided Masked Attention (FGMA) alignment mechanism to address the above issues.

Below, we present details of how we generate the aligned neighboring features by taking the alignment from $t-1$-th frame to $t$-th frame as an example. As shown in Fig. 2(b), firstly, we adopt SpyNet, a pretrained light-weight optical flow network, to calculate the bidirectional flow $\mathbf{O}_{t-1 \to t}$ and $\mathbf{O}_{t \to t-1}$. Secondly, We use the forward-backward consistency check to detect 'unreliable' regions where the flow-based alignment is likely to be inaccurate, denoted as mask $\mathbf{M}_{t-1 \to t}$. Specifically, we first warp $\mathbf{L}_t$ to $\mathbf{L}_{t-1}$ and obtain $\mathbf{L}_{t \to t-1}$, and then warp $\mathbf{L}_{t \to t-1}$ back to $\mathbf{L}_t$ and obtain $\mathbf{L}_{t \to t-1 \to t}$, i.e.,

$$
\begin{aligned}
\mathbf{L}_{t \to t-1} &= \mathit{Warp}\left(\mathbf{L}_t, \mathbf{O}_{t-1 \to t}\right), \\
\mathbf{L}_{t \to t-1 \to t} &= \mathit{Warp}\left(\mathbf{L}_{t \to t-1}, \mathbf{O}_{t \to t-1}\right).
\end{aligned}
\tag{4}
$$

The absolute difference between $\mathbf{L}_t$ and $\mathbf{L}_{t \to t-1 \to t}$ can quantify the inconsistency in bidirectional flow warping and serves as a direct measure of occlusion regions, i.e.,

$$
\mathbf{D}_{t-1 \to t}(i,j) = |\mathbf{L}_{t \to t-1 \to t}(i,j) - \mathbf{L}_t(i,j)|,
\tag{5}
$$

where $\mathbf{D}_{t-1 \to t}$ denotes the absolute difference map. Then, we introduce sensitivity factor $s$ to compute the occlusion mask, i.e.,

$$
\mathbf{M}_{t-1 \to t}(i,j) = \begin{cases} 1 & \text{if } s \cdot \mathbf{D}_{t-1 \to t}(i,j)/255 > 0.5, \\ 0 & \text{otherwise.} \end{cases}
\tag{6}
$$

With this mask obtained, we can identify the occluded regions where optical flow estimation may be unreliable. For these regions, we suggest employing attention mechanisms for alignment refinement. Therefore, we compute the query vector with this mask, and keys as well as values from the neighboring frame features, i.e.,

$$
\begin{aligned}
\mathbf{Q}_{t-1 \to t} &= Proj_q(\mathbf{F}_t^{in} \odot \mathbf{M}_{t-1 \to t}), \\
\mathbf{K}_{t-1 \to t} &= Proj_k(\mathbf{F}_{t-1}^i), \\
\mathbf{V}_{t-1 \to t} &= Proj_v(\mathbf{F}_{t-1}^i),
\end{aligned}
\tag{7}
$$

where $Proj_q$, $Proj_k$, and $Proj_v$ represent the point-wise convolution used for linear projections. Finally, the occlusion regions refined by attention are concatenated with the warped feature map $\mathbf{F}_{t-1 \to t}^{flow}$ and the occlusion mask $\mathbf{M}$, as the output of our FGMA module. i.e.,

$$
\begin{aligned}
\mathbf{F}_{t-1 \to t}^{att} &= Softmax\left(\frac{\mathbf{Q}_{t-1 \to t}\mathbf{K}_{t-1 \to t}^T}{\sqrt{d}}\right)\mathbf{V}_{t-1 \to t}, \\
\mathbf{F}_{t-1 \to t}^{flow} &= Warp\left(\mathbf{F}_{t-1}^i, \mathbf{O}_{t \to t-1}\right), \\
\mathbf{F}_{t-1 \to t}^{out} &= Concat\left(\mathbf{F}_{t-1 \to t}^{flow}, \mathbf{M}_{t-1 \to t}, \mathbf{F}_{t-1 \to t}^{att}\right).
\end{aligned}
\tag{8}
$$

To support dynamic adjustments in testing computational costs, we implemented three alignment branches during training: pure optical flow, pure attention, and our FGMA method. The percentage of non-zeros in the FGMA mask is dynamically controlled by the parameter $s$ in Eq. (6), enabling a continuous shift from optical flow to attention dominance. Therefore, we set four key boundaries: $s = 0$ (optical flow only), $s = 15$ (balancing flow and attention), $s = 100$ (attention-dominated), and $s = \infty$ (attention-only) along with other 12 sample points: six uniformly sampled from $s \in (0, 1]$ and six uniformly sampled from $s \in (1, 100)$. In this way, our experimental results form a characteristic performance curve as shown in Fig. 1. For DeAltHDR, the lower left point represents testing only with the optical flow branch, yielding suboptimal performance but at the lowest computational cost. The upper right point represents evaluation with the attention-only branch, which achieves the highest PSNR but at maximal computational complexity. The intermediate points along the curve represent evaluations using our FGMA branch with dynamically adjusted masks, and each point employs a different adjusted mask.

### 3.4 Self-Supervised Adaptation Method

It is still challenging to simulate realistic video sequences in alternating exposure patterns with real-world degradations such as different noise variations and motion blur. The inevitable domain gap between synthetic datasets and real-world LDR sequences limits the generalization capability of models trained only with synthetic data. As a result, they often produce artifacts during HDR video reconstruction. BracketIRE (Zhang et al., 2025) suggests a self-supervised fine-tuning method for real-world unlabeled HDR image reconstruction. However, it yields suboptimal performance when applied to video sequences with diverse motion. To this end, we propose to extend this with a motion-enhanced sampling strategy.

---

**Algorithm 1** Self-Supervised Adaptation Loss

---

**Require:** 5 neighboring frames $\mathcal{W} \leftarrow \{\mathbf{L}_i^c\}_{i=t-2}^{t+2}$
   EMA parameters: $\beta = 1.0$, $a = 0.999$
   Tone-mapping function $\mathcal{T}(\cdot)$ defined as:
   $\mathcal{T}(\mathbf{H}_t) = \frac{\log(1+\mu\mathbf{H}_t)}{\log(1+\mu)}$, where $\mu = 5000$
**Ensure:** Self-supervised adaptation loss $\mathcal{L}_{total}$
 1: {Frame selection rules}
 2: $\mathcal{G}_A \leftarrow \{\mathbf{L}_i^c\}_{i=t\pm2k}$, $k \in \{1,2,3\}$
 3: $\mathcal{G}_B \leftarrow \{\mathbf{L}_i^c\}_{i=t\pm(2k-1)}$, $k \in \{1,2,3\}$
 4: {Processing inputs}
 5: $\hat{\mathbf{H}}_t \leftarrow \mathcal{B}(\mathcal{W}; \Theta_{\mathcal{B}})$
 6: {Dynamic subset construction}
 7: $\mathbf{L}_A^c \leftarrow \text{RandomSelectOne}(\mathcal{G}_A)$
 8: $\mathbf{L}_B^c \leftarrow \text{RandomSelectOne}(\mathcal{G}_B)$
 9: $\mathcal{S} \leftarrow \{\mathbf{L}_t^c, \mathbf{L}_A^c, \mathbf{L}_B^c\}$
10: {Loss computation}
11: $\tilde{\mathbf{H}}_t \leftarrow \mathcal{B}(\mathcal{S}; \Theta_{\mathcal{B}})$
12: $\mathcal{L}_{\text{time}} \leftarrow \|\mathcal{T}(\tilde{\mathbf{H}}_t) - \mathcal{T}(\text{sg}(\hat{\mathbf{H}}_t))\|_1$ {Temporal loss}
13: $\Theta_{\mathcal{B}_k}^{\text{ema}} \leftarrow a\Theta_{\mathcal{B}_{k-1}}^{\text{ema}} + (1-a)\Theta_{\mathcal{B}_k}$
14: $\hat{\mathbf{H}}_t^{\text{ema}} \leftarrow \mathcal{B}(\mathcal{W}; \Theta_{\mathcal{B}}^{\text{ema}})$
15: $\mathcal{L}_{\text{ema}} \leftarrow \|\mathcal{T}(\tilde{\mathbf{H}}_t) - \mathcal{T}(\text{sg}(\hat{\mathbf{H}}_t^{\text{ema}}))\|_1$ {EMA loss}
16: $\mathcal{L}_{\text{total}} \leftarrow \mathcal{L}_{\text{time}} + \beta\mathcal{L}_{\text{ema}}$ {Total loss}
17: **return** $\mathcal{L}_{total}$

---

Specifically, the input of the model is a sequence of 5 consecutive frames $\{\mathbf{L}_i^c\}_{i=t-2}^{t+2}$ where $\mathbf{L}_t^c$ is the current frame. The output of these 5 frames is $\hat{\mathbf{H}}_t$. Then, we create a 3-frame subset for supervision by: (1) always including the current frame $L_t$ as the target, (2) randomly selecting one long-exposure neighboring frame and (3) randomly selecting one short-exposure neighboring frame. The output of these 3 frames is denoted as $\tilde{\mathbf{H}}_t$. Generally, $\hat{\mathbf{H}}_t$ performs better than $\tilde{\mathbf{H}}_t$. Therefore, although no ground-truth is provided, $\hat{\mathbf{H}}_t$ can be seen as the pseudo-target of $\tilde{\mathbf{H}}_t$. This random sampling strategy introduces inter-frame motion diversity, improving temporal consistency across reconstructed frames. Moreover, we follow BracketIRE to use an exponential moving average (EMA) regulation loss, which can stabilize the training process. The detailed loss is summarized in Algorithm 1.

## 4 Datasets

### 4.1 Synthetic Dataset

To synthesize more realistic LDR alter-exposure video sequences, we need to minimize the gap between synthetic and real frames. First, we capture high-quality HDR 4K/60fps video with a DJI Pocket 3 gimbal camera, ensuring frame-by-frame clarity during high-motion scenarios by its 3-axis mechanical stabilization system. Furthermore, its 1-inch large sensor significantly enhances low-light performance and captures richer details. Next, following the approach of (Nah et al., 2019) to perform frame interpolation. We adopt RIFE (Huang et al., 2022), a pre-trained video frame interpolation model, to increase the frame rate by 16 times and get the HDR sequence $\{\mathbf{H}_t\}_{t=1}^{T}$ Finally, we introduce our degradation method to simulate alter-exposure sequences. Step 1: the input sequence is divided into non-overlapping 64-frame chunks. Step 2: in each group, the first frame is used to simulate the short-exposure frame, while the remaining 63 frames are used to simulate the long-exposure frame. Step 3: the HDR images are then converted to LDR by value-clipping and 10-bit linear quantization. Step 4: we add heteroscedastic Gaussian noise (Brooks et al., 2019; Wang et al., 2020; Hasinoff et al., 2010) to the raw version of the sequence (which is generated using UPI (Brooks et al., 2019)), and adjust the brightness to match the exposure values

Table 1: Quantitative comparison with state-of-the-art HDR restoration methods on both synthetic and real-world datasets. The best results are **bolded**, and the second-best results are underlined.

| | Methods | | Synthetic | | | Real-World | |
|---|---|---|---|---|---|---|---|
| | | PSNR↑ | SSIM↑ | LPIPS↓ | HDR-VDP-2↑ | CLIPIQA↑ | MANIQA↑ |
| HDR Image | AHDRNet (CVPR 19) | 31.57 | 0.9588 | 0.226 | 68.24 | 0.2032 | 0.2098 |
| | SCTNet (ICCV 23) | 31.95 | 0.9618 | 0.205 | 73.34 | 0.2320 | 0.2492 |
| | SAFNet (ECCV 24) | 32.02 | 0.9619 | 0.202 | 72.96 | 0.2423 | 0.2502 |
| | BracketIRE (ICLR 25) | 32.17 | 0.9623 | 0.200 | 75.32 | 0.2584 | 0.2692 |
| HDR Video | Chen et al. (ICCV 21) | 31.98 | 0.9612 | 0.208 | 75.67 | 0.2356 | 0.2472 |
| | LAN-HDR (ICCV 23) | 32.04 | 0.9614 | 0.211 | 76.02 | 0.2546 | 0.2634 |
| | NECHDR (ACM MM 24) | 32.16 | 0.9619 | 0.205 | 75.42 | 0.2578 | 0.2706 |
| | HDRFlow (CVPR 24) | 32.26 | 0.9629 | 0.196 | 76.56 | 0.2601 | 0.2694 |
| DeAltHDR | w/o Adaptation ($s = 15$) | **32.55** | **0.9644** | **0.192** | **77.02** | 0.2621 | 0.2734 |
| | w/ Adaptation | - | - | - | - | **0.2679** | **0.2774** |

defined in Section 3.1. The noise variance depends on the intensity of the pixels, with parameters estimated from captured real-world images. In total, we collected 200 scenes (100 daytime scenes / 100 nighttime scenes), including 4000 data pairs (200 data pairs in each scene). 176 scenes (88 daytime scenes / 88 nighttime scenes) are used for training, while the remaining 24 scenes (12 daytime scenes / 12 nighttime scenes) are used for testing. More details on the comparison with other datasets are provided in Appendix A.

## 4.2 REAL-WORLD DATASET

To construct a realistic HDR video dataset with real noise and motion blur, we used an iPhone 16 Pro Max with the ProShot App. This App is configured to capture alternating-exposure pairs (EV-2 and EV+1) in both RAW (DNG) and RGB format, with a fixed ISO. It is worth noticing that we intentionally shake the iPhone slightly to preserve natural motion blur. Moreover, global motion (only the camera is moving), local motion (only the foreground is moving), and full motion (both the camera and foreground are moving) scenes are collected in both daytime and nighttime. Finally, we collected 100 alter-exposed video sequences, each of them contains 100 frames. 80 scenes are used for training, and the remaining 20 scenes are used for testing. More details of comparison with other datasets is written in Appendix B.

## 5 EXPERIMENTS

### 5.1 EXPERIMENT SETUP

**Network Details** Following the settings of Turtle (Ghasemabadi et al., 2024), we adopt a 5-frame input structure with specialized dual-encoder processing. We employ $\ell_1$ loss and VGG perceptual loss to optimize the network, which can be defined as,

$$
\begin{aligned}
\mathcal{L}_1 &= \|\mathcal{T}(\hat{\mathbf{H}}_{\mathbf{t}}) - \mathcal{T}(\mathbf{H}_{\mathbf{t}}^{\mathbf{gt}})\|_1, \\
\mathcal{L}_{vgg} &= \sum_i \|\phi_i(\mathcal{T}(\hat{\mathbf{H}}_{\mathbf{t}})) - \phi_i(\mathcal{T}(\mathbf{H}_{\mathbf{t}}^{\mathbf{gt}}))\|_1,
\end{aligned}
\tag{9}
$$

where $\hat{\mathbf{H}}_{\mathbf{t}}$ is the t-th frame and $\mathbf{H}_{\mathbf{t}}^{\mathbf{gt}}$ is the t-th ground truth frame. $\phi_i(\cdot)$ denotes the feature extractor from the i-th layer of VGG16 network. Specifically, the total loss can be defined as

$$
\mathcal{L}_{total} = \mathcal{L}_1 + \lambda_{vgg}\mathcal{L}_{vgg}.
\tag{10}
$$

**Implementation details** To enable evaluation with dynamically adjustable computational costs, we implement a hybrid alignment strategy. Specifically, 30% of the training batches utilize optical flow-based alignment, while another 30% employ implicit attention mechanisms. The remaining 40% batches use our proposed **Flow-Guided Mask Attention Alignment**, where the mask size is randomly determined by adjusting the threshold parameter $s$ in Eq. (6). We implemented our network in PyTorch (Paszke et al., 2019) and conducted experiments on a single NVIDIA RTX A6000 (48GB) GPU. The batch size is set to 8 and the input patch size is set to $192 \times 192$. We adopt

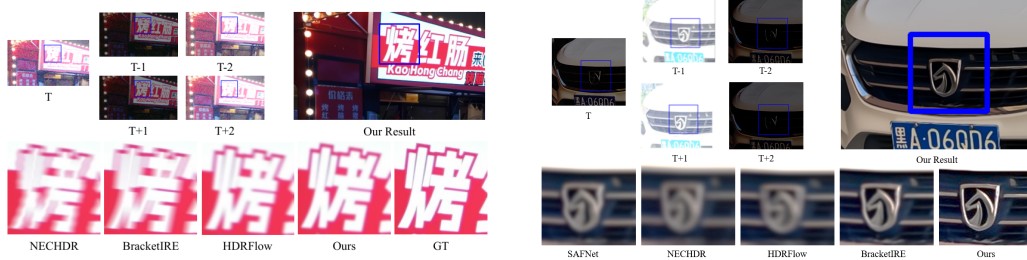

Figure 3: Comparisons on synthetic dataset.  Figure 4: Comparisons on real-world dataset.

Table 2: Temporal consistency evaluation on the synthetic dataset.

| Method | PSNR↑ | TWE↓ | TLP↓ | TOF↓ |
|---|---|---|---|---|
| HDRFlow | 32.26 | 2.84 | 2.8756 | 4.0238 |
| NECHDR | 32.16 | 2.91 | 3.0486 | 4.3642 |
| DeAltHDR | **32.55** | **2.68** | **2.7142** | **3.2117** |

Table 3: Comparison of computational costs.

| Methods | FLOPs (G) | Time (ms) |
|---|---|---|
| AHDRNet (CVPR 19) | 146 | 140 |
| SCTNet (ICCV 23) | 338 | 356 |
| SAFNet (ECCV 24) | 268 | 290 |
| BracketIRE (ICLR 25) | 382 | 387 |
| Chen et al. (ICCV 21) | 282 | 302 |
| LAN-HDR (ICCV 23) | 332 | 325 |
| NECHDR (ACM MM 24) | 296 | 320 |
| HDRFlow (CVPR 24) | 116 | **128** |
| DeAltHDR ($s = 15$) | 128 | 152 |

Table 4: Effect of alignment methods.

| Alignment Methods | PSNR↑ | FLOPs (G) | Time (ms) |
|---|---|---|---|
| Flow-Guided Defor. Conv. | 32.42 | 102 | 112 |
| Guided Defor. Attention | 32.46 | 202 | 244 |
| Patch Alignment | 32.41 | 178 | 198 |
| DeAltHDR ($s = 0$) | 32.42 | **84** | **94** |
| DeAltHDR ($s = 0.71$) | 32.49 | 115 | 134 |
| DeAltHDR ($s = 15$) | 32.55 | 128 | 152 |
| DeAltHDR ($s = 100$) | 32.63 | 159 | 198 |
| DeAltHDR ($s = \infty$) | **32.65** | 169 | 224 |

the AdamW optimizer (Loshchilov & Hutter, 2017) with $\beta_1 = 0.9$ and $\beta_2 = 0.999$. The models are trained for 250 epochs on the synthetic dataset and 20 epochs on the real-world dataset, with an initial learning rate of $4e^{-4}$ and $1e^{-6}$, respectively. We use the cosine annealing strategy (Loshchilov & Hutter, 2016) to decrease the learning rates to $1e^{-7}$. $\lambda_{vgg}$ is 0.5.

## 5.2 EVALUATIONS AND COMPARISON CONFIGURATIONS

**Evaluation Configurations** For the synthetic dataset, we adopt PSNR, SSIM (Wang et al., 2004), LPIPS (Zhang et al., 2018) and HDR-VDP-2 (Mantiuk et al., 2011) as evaluation metrics. The PSNR, SSIM and LPIPS are computed in the $\mu$-law tonemapped domain. For the real-world ones, we use no-reference metrics CLIPIQA (Wang et al., 2023) and MANIQA (Yang et al., 2022) as evaluation metrics in the absence of ground-truth HDR images. Moreover, we conducted additional experiments to evaluate temporal consistency with three widely-used metrics in video processing: tOF (Chu et al., 2020), which measures the pixel-wise difference in motion es-timated from sequences; tLP (Chu et al., 2020), which assesses perceptual changes over time using deep feature maps; and Temporal Warping Error(TWE), which quantifies frame-to-frame consistency following motion compensation.

**Comparison Configurations** We compare our method with state-of-the-art approaches, including HDR image restoration methods (*i.e.* AHDRNet (Yan et al., 2019), SCTNet (Tel et al., 2023b), SAFNet (Kong et al., 2024), and BracketIRE (Zhang et al., 2025)) and HDR video restoration methods (*i.e.* Chen *et al.* (Chen et al., 2021), LAN-HDR (Chung & Cho, 2023a), NECHDR (Cui et al., 2024), and HDRFlow (Xu et al., 2024)). For evaluation on synthetic images, we use our synthetic dataset to train all these methods. For evaluation on real-world images, we use the original pre-trained models released by these methods. while our models are further adapted to real-world data by the proposed self-supervised adaptation method. Furthermore, we also conducted experiments on other datasets in Appendix G.

Table 5: Ablation study on encoder parameter sharing strategies. "✓" indicates parameters are independent between long and short exposures, while "✗" indicates shared parameters. Best results are in **bold**.

| Level 1 | Level 2 | Level 3 | PSNR↑ | SSIM↑ | LPIPS↓ | HDR-VDP-2↑ |
|---------|---------|---------|-------|-------|--------|------------|
| ✓ | ✓ | ✓ | **32.55** | **0.9644** | **0.192** | **77.02** |
| ✓ | ✓ | ✗ | 32.40 | 0.9623 | 0.195 | 76.52 |
| ✓ | ✗ | ✗ | 32.18 | 0.9598 | 0.204 | 75.56 |
| ✗ | ✗ | ✗ | 31.96 | 0.9587 | 0.211 | 74.78 |

Table 6: Effect of self-supervised adaptation methods.

| Self-Supervised Methods | CLIPIQA↑ | MANIQA↑ |
|-------------------------|----------|---------|
| w/o Adaptation | 0.2621 | 0.2734 |
| TMRNet | 0.2648 | 0.2732 |
| Ours | **0.2679** | **0.2774** |

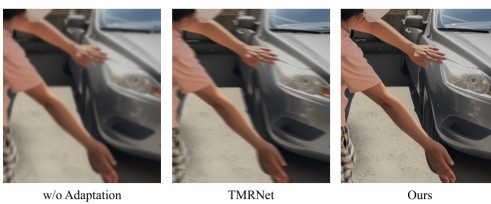

Figure 5: Effect of adaptation methods.

## 5.3 EXPERIMENT RESULTS

**Results on Synthetic Dataset** As shown in Table 1, the middle four columns report full-reference evaluation results on the synthetic dataset, including PSNR, SSIM, LPIPS, and HDR-VDP-2. Our method outperforms all competing HDR image and video reconstruction methods across these metrics. These improvements validate the effectiveness of our flow-guided masked attention alignment strategy. As visualized in Fig. 3, our method restores more structure and reduces artifacts more effectively than previous approaches. Moreover, as shown in Table 2, our DeAltHDR outperforms both NECHDR (Cui et al., 2024) and HDRFlow (Xu et al., 2024) across all three metrics. This means that our DeAltHDR not only performs well on single images, but also has better temporal consistency and the generated videos are much smoother and have less flickering.

**Results on Real-World Dataset** The last two columns of Table 1 present no-reference evaluation results on real-world data, using CLIP-IQA and MANIQA. Even trained solely on synthetic data (*i.e.* w/o Adaptation), DeAltHDR outperforms existing methods when directly evaluated on real-world datasets. This demonstrates the strong generalization ability of our model design. Furthermore, by applying our proposed self-supervised adaptation strategy (*i.e.* w/ Adaptation), DeAltHDR achieves further improvements. As shown in Fig. 4, models trained on synthetic datasets tend to produce visible artifacts when applied to real-world data. By introducing our self-supervised adaptation method, these artifacts are effectively reduced, leading to improved visual quality in real-world scenarios. More results can be found in Appendix G.

**Computational Efficiency** As shown in Table 3, DeAltHDR achieves competitive computational efficiency and has a comparable inference time to HDRFlow, currently the fastest baseline. Compared to other state-of-the-art methods, it is significantly more efficient in both computational cost and runtime. In summary, our method not only achieves better reconstruction performance, but also maintains high efficiency. More details can be found in Appendix E.

## 5.4 ABLATION STUDY

**Effect of Flow-Guided Masked Attention Alignment** To validate the effectiveness of our proposed Flow-Guided Masked Attention (FGMA), we replace the alignment module in DeAltHDR with several existing alternatives. Specifically, we adopt guided deformable attention from RVRT (Liang et al., 2022), patch alignment from PSRT-Recurrent (Shi et al., 2022) and flow-guided deformable convolution from BasicVSR++ (Chan et al., 2022). As shown in Table 4, FGMA equipped DeAltHDR not only achieves a higher PSNR but also significantly reduces computational cost, demonstrating the effectiveness of FGMA in balancing reconstruction quality and efficiency.

**Effect of dual-encoder processing** We also discuss the role of the dual-encoder design. We conducted an ablation study with four different training configurations: 1) using independent parameters

at all three levels (no sharing), 2) sharing parameters across all three levels, 3) sharing parameters only at level 3, and 4) sharing parameters at levels 2 and 3 but not at level 1.

As shown in Table 5, the best performance is achieved when no parameters are shared across the three levels. This result can be attributed to the inherent characteristics of the input data: short-exposure frames exhibit higher noise levels, while long-exposure frames suffer from more severe motion blur. Therefore, using shared parameters for the encoders, which process these fundamentally different inputs, is suboptimal. As a result, independent encoders are more effective, as they specialize in extracting features from their respective inputs with distinct artifacts.

**Effect of Self-Supervised Adaptation** To evaluate the effectiveness of our self-supervised adaptation strategy, we replace the fine-tuning scheme in DeAltHDR with that of TMRNet (Zhang et al., 2025). TMRNet also adopts a multi-frame self-supervised loss, but its training frames are limited to a strict subset of the input sequence. As a result, it limits performance in video HDR reconstruction, where large motion ranges are common. Instead, we propose to extend the frame sampling range from $t - 2$ to $t + 2$ (5 frames) to $t - 6$ to $t + 6$ (13 frames) to capture larger motion ranges. To maintain the dynamic range of sparse frames, we randomly sample one short-exposure frame and one long-exposure frame from this extended range. As shown in Table 6, our self-supervised adaptation method achieves better performance in both CLIP-IQA and MANIQA compared to TMRNet and the baseline (*i.e.* w/o Adaptation). As shown in Fig. 5, our method is able to reconstruct more accurate HDR videos with finer detail preservation and fewer artifacts.

## 6    CONCLUSION

In this paper, we propose DeAltHDR, a robust framework for high-quality HDR video reconstruction from degraded alternating-exposure sequences. Our approach introduces three key innovations: (1) a Flow-Guided Masked Attention(FGMA) mechanism that dynamically combines optical flow and sparse attention for efficient alignment under noise and blur, (2) a motion-enhanced self-supervised adaptation method for effective real-world fine-tuning, and (3) comprehensive synthetic and real-world datasets that capture authentic noise and motion blur characteristics. Extensive experiments demonstrate that DeAltHDR outperforms state-of-the-art methods in both synthetic and real-world datasets due to our FGMA module and self-supervised method. To the best of our knowledge, DeAltHDR is the first framework that addresses noise, blur, and motion challenges in alternating-exposure HDR video reconstruction.

## 7    APPLICATIONS AND LIMITATIONS

**Applications.** A primary application of this work is high-quality HDR video capture in challenging lighting conditions, particularly for dynamic nighttime scenes. Our method enables smartphones and consumer cameras to produce HDR videos that preserve both bright highlights and dark details while remaining free from noise and motion blur artifacts. This capability is valuable for various practical scenarios, including mobile cinematography, live streaming, surveillance systems, and documentary filmmaking in low-light environments. We demonstrate its effectiveness on real-world data captured with commodity devices (*i.e.*, iPhone 16 Pro Max), showing that our approach is readily deployable on consumer hardware. Moreover, the adjustable computational cost feature allows users to balance quality and speed based on their specific needs, making it suitable for both real-time applications and offline processing.

**Limitations.** Despite its effectiveness, our method has several limitations. First, it requires alternating-exposure input sequences, which limits its applicability to videos captured with standard fixed-exposure settings. Second, it may fail in extreme cases where all input frames lack sharp references due to severe camera shake or fast motion, as the multi-frame fusion mechanism relies on complementary sharp information. Third, given the diverse noise characteristics of different camera sensors, our model trained on one sensor may exhibit limited generalization when applied to sensors with significantly different noise profiles. Finally, although our method achieves competitive efficiency, the computational cost remains non-trivial for real-time processing on resource-constrained devices, particularly when using higher attention ratios. We leave the investigation of a more general, sensor-agnostic model for future work.

ACKNOWLEDGMENTS

This work was supported by the National Natural Science Foundation of China (NSFC) under Grant No. 62476067, and the China Postdoctoral Science Foundation under Grant No. 2025M784371.

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

# APPENDIX

The content of the appendix involves:

## A  DETAILS OF SYNTHETIC DATASET

Fig. A provides the overview of our synthetic data generation pipeline. The original HDR videos in our synthetic dataset were captured with the DJI Pocket 3. First, we used the UPI method (Brooks et al., 2019) to convert these RGB videos to RAW space to add noise and convert them back to RGB space. Unlike the default parameters in UPI, our method employs the real camera parameters from the DJI Pocket 3 for both the RGB-to-RAW and RAW-to-RGB conversions. Second, we employ RIFE (Huang et al., 2022) for frame interpolation. We confirm that RIFE is agnostic to dynamic-range encoding of the input data and can be employed for HDR video interpolation. First, RIFE is fed HDR frames that are normalized float tensors in the [0, 1] range. Since RIFE operates internally on floating-point data, this linear scaling preserves high dynamic range information and relative luminance without the quantization issue. Second, the core objective of RIFE is motion estimation and occlusion-aware blending at intermediate-time ($t = 0.5$). Motion estimation is fundamentally geometric and largely invariant to monotonic intensity transformations (linear scaling, $\gamma$-correction or $\mu$-law). Therefore, scaling HDR data to $[0, 1]$ ensures numerical stability without distorting the underlying motion field. Unlike classical brightness-constancy methods that rely on rigid photometric models, RIFE learns flow and fusion weights end-to-end in a normalized feature space. In this way, RIFE is agnostic to dynamic-range encoding of the input data. Visual results can be referred to Figure K. While it preserves the data distribution as confirmed by visual inspection, the synthesized motion blur still diverges from real-world blur due to its inherent limitations. To address this gap, we propose a self-supervised real-world adaptation approach in this work. Given recent advances in interpolation models for large motions (Jain et al., 2024) and complex textures (Zhong et al., 2024), we believe that this limitation will be progressively mitigated as these technologies continue to advance.

The noise in RAW images consists primarily of two components: shot noise and read noise (Brooks et al., 2019). The shot noise follows a Poisson distribution with a mean corresponding to the actual light intensity in photoelectrons, while the read noise can be modeled as a zero-mean Gaussian random variable with constant variance. These noise sources can be jointly approximated as a heteroskedastic Gaussian random variable $\mathbf{N}$, which can be defined as:

$$\mathbf{N} \sim \mathcal{N}(\mathbf{0}, \lambda_{read} + \lambda_{shot}\mathbf{X}), \tag{A}$$

where $\mathbf{X}$ represents the clean signal intensity and the parameters $\lambda_{read}$ and $\lambda_{shot}$ are determined by the sensor's analog and digital gain settings.

To ensure realistic noise synthesis, we calibrate our model using the noise characteristics of the iPhone 16 Pro Max camera sensor, extracting $\lambda_{shot}$ and $\lambda_{read}$ from the raw image metadata. The ISO of the reference images is set to 100 during daytime and 1600 during nighttime. For these images, we measured $\lambda_{shot}^{day} \approx 3.23 \times 10^{-4}$, $\lambda_{shot}^{night} \approx 4.52 \times 10^{-3}$ and $\lambda_{read}^{day} \approx 2.67 \times 10^{-6}$, $\lambda_{read}^{night} \approx$

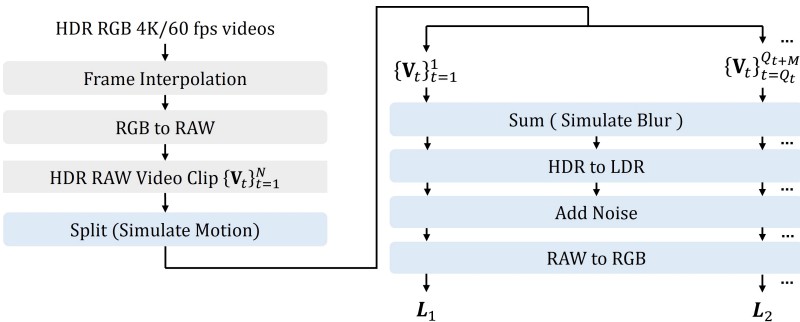

Figure A: Overview of our synthetic data generation pipeline. We utilize HDR video to synthesize multi-exposure images $\{\mathbf{L}_t\}_{t=1}^{N}$. $Q_t$ marks the initial frame, and $M$ specifies the frame numbers used to synthesize long-exposure frames. The short-exposure ground truth is its corresponding HDR frame. The long-exposure ground truth is the middle frame used to simulate this frame sequence.

$4.04 \times 10^{-5}$. To generate noise with different intensity levels, we sample these parameters uniformly across the ISO range 50-400 at daytime and 800-3200 at nighttime, leading to the following distributions:

$$\log(\lambda_{shot}^{day}) \sim \mathcal{U}(\log(0.00014), \log(0.00066)),$$
$$\log(\lambda_{shot}^{night}) \sim \mathcal{U}(\log(0.0021), \log(0.0086)),$$
$$\log(\lambda_{read}^{day} \mid \log(\lambda_{shot}^{day})) \sim$$
$$\mathcal{N}(1.428 \log(\lambda_{shot}^{day}) + 1.215, 0.28^2),$$
$$\log(\lambda_{read}^{night} \mid \log(\lambda_{shot}^{night})) \sim$$
$$\mathcal{N}(1.692 \log(\lambda_{shot}^{night}) + 0.418, 0.25^2),$$

(B)

$\mathcal{U}(a, b)$ denotes the uniform distribution over interval $[a, b]$.

Table A: Comparative analysis of key characteristics in HDR video datasets. NLE and CPBD quantify the challenging artifacts (noise and blur) in the input LDR sequences, whereas DR, SI, and stdL describe the richness and quality of the HDR ground truth.

| Datasets | Noise | Blur | NLE↓ | CPBD↑ | DR↑ | SI↑ | stdL↑ |
|---|---|---|---|---|---|---|---|
| Cinematic Video | ✓ | ✗ | 18.4 | 0.65 | 2.46 | 9.02 | 11.26 |
| DeepHDRVideo | ✗ | ✗ | 6.6 | 0.79 | 2.54 | 8.93 | 11.35 |
| Real-HDRV | ✓ | ✗ | 13.5 | 0.40 | 2.73 | 9.16 | 12.13 |
| Our Synthetic Dataset | ✓ | ✓ | 28.5 | 0.28 | 2.80 | 9.13 | 12.42 |
| Our Real-World Dataset | ✓ | ✓ | 24.6 | 0.32 | - | - | - |

## B    COMPARISON WITH OTHER DATASETS

In this section, we present a comparative analysis between our dataset and existing datasets. Currently, DeepHDRVideo (Chen et al., 2021) and Cinematic Video (Froehlich et al., 2014) are among the most widely adopted datasets for HDR reconstruction of single images and videos. Recently, (Shu et al., 2024) presented a new large-scale real-world dataset named Real-HDRV, which contains various scenes and diverse motion patterns. The following comparisons will focus on these three key datasets.

A key distinction is that our work explicitly accounts for both noise and blur artifacts introduced during real-world capture processes. Notably, the DeepHDRVideo dataset only accounts for luminance variations between long- and short-exposure frames, but fails to incorporate realistic noise and blur degradations. The Cinematic Video dataset is synthesized through simulation of HDR video. The source HDR videos used for the simulation inherently contain motion blur. Both the synthesized

Table B: Scene-wise dynamic range statistics for our synthetic dataset.

| Scene Type | Scenes | Avg DR | Min (cd/m$^2$) | Max (cd/m$^2$) |
|---|---|---|---|---|
| Daytime Indoor | 16 | 2.46 | 0.08 | $3.2 \times 10^2$ |
| Daytime Outdoor | 34 | 2.82 | 0.15 | $6.0 \times 10^3$ |
| Nighttime Indoor | 14 | 2.32 | 0.02 | $8.0 \times 10^1$ |
| Nighttime Outdoor | 36 | 3.24 | $5 \times 10^{-3}$ | $1.2 \times 10^3$ |
| Overall | 100 | 2.71 | $5 \times 10^{-3}$ | $6.0 \times 10^3$ |

Table C: Component-wise effect on degradation handling. Short-exposure metrics highlight noise suppression (PSNR/NLE↓), while long-exposure metrics focus on blur fidelity (PSNR/CPBD↑).

| Variant | Short Exposure (noise) | | Long Exposure (blur) | |
|---|---|---|---|---|
| | PSNR↑ | NLE↓ | PSNR↑ | CPBD↑ |
| DeAltHDR ($s$=15) | **32.65** | **2.92** | **32.45** | **0.75** |
| w/o Dual Encoders | 32.04 | 7.27 | 31.88 | 0.51 |
| Flow-only alignment ($s$=0) | 32.48 | 4.05 | 32.36 | 0.64 |
| Attention-only alignment ($s=\infty$) | 32.74 | 2.23 | 32.56 | 0.79 |
| w/o VGG perceptual loss | 32.46 | 4.11 | 32.38 | 0.60 |

long- and short-exposure frames simply inherit this blur, without considering the difference in motion blur between long and short exposures. In real-world scenes, short-exposure images should have less blur than long-exposure images. The recently proposed Real-HDRV dataset still mainly considers luminance variations, exhibiting less noise and blur. Our synthetic dataset utilizes 4K/60fps HDR videos captured exclusively with DJI Pocket 3, ensuring per-frame sharpness while simulating realistic overexposure and motion blur effects through multi-frame superposition, and each frame has its corresponding ground truth (GT) image. For real-world data acquisition, we employed an iPhone 16 Pro Max with ProShot's bracketing mode while disabling all noise reduction modules to preserve authentic sensor noise characteristics. At the same time, camera shake and object motion enable the generation of physically accurate motion blur in long-exposure frames. Therefore, our dataset is more comprehensive and physically accurate than existing alternatives by incorporating real-world noise and motion blur.

Table A presents the quantitative statistical results for each dataset. NLE (Liu et al., 2012) measures the intensity of image noise, where a lower value indicates less noise, and vise versa. CPBD (Narvekar & Karam, 2011) measures the degree of motion blur in images, where a lower value indicates more blur and vise versa. In particular, we evaluate NLE only on the short-exposure images and CPBD only on the long-exposure images. DR (Hu et al., 2022) is calculated as the log10 difference between the highest 2% luminance and the lowest 2% luminance. SI (Spatial Information) is defined in (BT, 2020), and stdL (standard deviation of luminance) is defined in (Guo et al., 2023). As shown in Table A, our dataset exhibits significant levels of noise and motion blur in the LDR inputs. Moreover, the inclusion of SI, DR, and stdL metrics provides comprehensive characterization of spatial complexity, luminance range diversity, and contrast richness in HDR ground truth images. Furthermore, we provide comprehensive dataset statistics and visualizations in Table B, and more visualization results can be found in Figure F and Figure G.

## C  DETAILS OF OUR ENCODER AND DECODER BLOCK

Our network adopts a U-net architecture with skip connections, featuring a three-level encoder and a three-level decoder. The first two levels of the encoder use identical block designs, differing only in the number of blocks and feature resolution. Each level consists of multiple ReducedAttn blocks with standard feedforward networks (FFW). The ReducedAttn module replaces conventional attention mechanisms with a more efficient design: it first expands channels using $1 \times 1$ convolutions, then applies depthwise $3 \times 3$ convolutions for spatial mixing, followed by another $1 \times 1$ convolution to project features back. Each FFW component uses two $1 \times 1$ convolutions with a GELU activation in between to mix channel information. Level 3 of the encoder differs from the first two levels, featuring a deeper architecture with six blocks. While levels 1–2 use ReducedAttn, level 3 employs

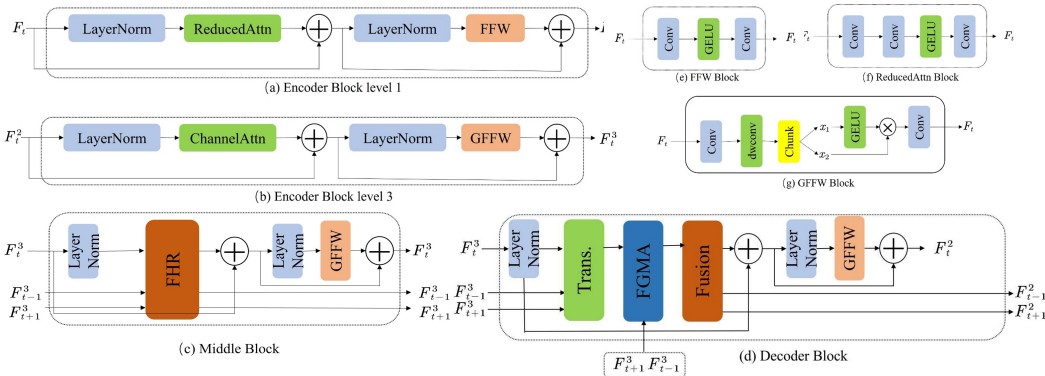

Figure B: Details of our network architecture. (a) denotes the block used in the first and second levels of the encoder. (b) shows the block used in the third level of the encoder. (c) denotes the middle block. (d) denotes the decoder block used in all three levels. (e) shows the FFW block. (f) shows the ReducedAttn block. (g) shows the GFFW block. Specifically, the number of blocks in each level is 2, 4, and 6, while the number of middle blocks is set to 8.

Table D: Short-exposure evaluation results.

|  | Methods | Synthetic (Short) | | | | Real-World (Short) | |
|---|---|---|---|---|---|---|---|
|  |  | PSNR↑ | SSIM↑ | LPIPS↓ | HDR-VDP-2↑ | CLIPIQA↑ | MANIQA↑ |
| HDR Image | AHDRNet (CVPR 19) | 31.68 | 0.9595 | 0.2247 | 68.51 | 0.2055 | 0.2117 |
|  | SCTNet (ICCV 23) | 32.08 | 0.9624 | 0.2039 | 73.58 | 0.2341 | 0.2509 |
|  | SAFNet (ECCV 24) | 32.11 | 0.9624 | 0.2010 | 73.19 | 0.2445 | 0.2518 |
|  | BracketIRE (ICLR 25) | 32.29 | 0.9629 | 0.1988 | 75.60 | 0.2604 | 0.2710 |
| HDR Video | Chen et al. (ICCV 21) | 32.08 | 0.9618 | 0.2069 | 75.93 | 0.2378 | 0.2489 |
|  | LAN-HDR (ICCV 23) | 32.15 | 0.9619 | 0.2098 | 76.26 | 0.2569 | 0.2653 |
|  | NECHDR (ACM MM 24) | 32.25 | 0.9625 | 0.2040 | 75.65 | 0.2599 | 0.2724 |
|  | HDRFlow (CVPR 24) | 32.34 | 0.9634 | 0.1951 | 76.78 | 0.2621 | 0.2711 |
| DeAltHDR | w/o Adaptation ($s$=15) | **32.65** | **0.9650** | **0.1911** | **77.29** | 0.2643 | 0.2752 |
|  | w/ Adaptation | – | – | – | – | **0.2700** | **0.2791** |

standard Channel Attention blocks in all layers, enabling global feature interactions. The FFW networks are upgraded to Gated FFW blocks, which introduces a gating mechanism through parallel depth-wise convolutions and element-wise multiplication before feature projection.

The Fusion module starts the middle block by applying channel-wise cross-attention layers. These layers maintain temporal coherence through efficient key-value buffers that cache and retrieve features from neighboring frames, creating a stable foundation for motion-aware processing. All decoder levels share the same fundamental block structure. In our network, we use the FGMA alignment method to align neighboring features, followed by the Fusion module, which is the same as that in the middle block. The detailed architecture is shown in Fig. B.

# D   DISCUSSION ON THE MODEL'S CAPABILITY FOR DENOISING AND DEBLURRING

In this section, we provide a detailed analysis of our model's capability for denoising and deblurring.

**Which Component Handles Noise and Blur?** The model inherently handles noise and blur by leveraging complementary information from multiple input frames. In addition, we conducted a comprehensive component-wise ablation study to investigate which component contributes the most. As shown in Table C, dual encoders contribute the most by extracting features from inputs with different degradations.

**Why Does the Model Handles Blur Without Explicit Deblurring Design** Our model successfully handles motion blur through two synergistic, implicit mechanisms that arise from its core design for video HDR.

Table E: Long-exposure evaluation results.

| | Methods | Synthetic (Long) | | | | Real-World (Long) | |
|---|---|---|---|---|---|---|---|
| | | PSNR↑ | SSIM↑ | LPIPS↓ | HDR-VDP-2↑ | CLIPIQA↑ | MANIQA↑ |
| HDR Image | AHDRNet (CVPR 19) | 31.46 | 0.9581 | 0.2273 | 67.97 | 0.2009 | 0.2079 |
| | SCTNet (ICCV 23) | 31.82 | 0.9612 | 0.2061 | 73.10 | 0.2299 | 0.2475 |
| | SAFNet (ECCV 24) | 31.93 | 0.9614 | 0.2030 | 72.73 | 0.2401 | 0.2486 |
| | BracketIRE (ICLR 25) | 32.05 | 0.9617 | 0.2012 | 75.04 | 0.2564 | 0.2674 |
| HDR Video | Chen et al. (ICCV 21) | 31.88 | 0.9606 | 0.2091 | 75.41 | 0.2334 | 0.2455 |
| | LAN-HDR (ICCV 23) | 31.93 | 0.9609 | 0.2122 | 75.78 | 0.2523 | 0.2615 |
| | NECHDR (ACM MM 24) | 32.07 | 0.9613 | 0.2060 | 75.19 | 0.2557 | 0.2688 |
| | HDRFlow (CVPR 24) | 32.18 | 0.9624 | 0.1969 | 76.34 | 0.2581 | 0.2677 |
| DeAltHDR | w/o Adaptation ($s=15$) | **32.45** | **0.9638** | **0.1929** | **76.75** | 0.2599 | 0.2716 |
| | w/ Adaptation | – | – | – | – | **0.2658** | **0.2757** |

Table F: Runtime and performance comparison of flow-based and attention-based alignment methods. Besides the default model, we develop a compact DeAltHDR-S variant whose inference time is lower than HDRFlow while maintaining quantitative results.

| Method | PSNR↑ | SSIM↑ | LPIPS↓ | HDR-VDP-2↑ | FLOPs (G) | Time (ms) |
|---|---|---|---|---|---|---|
| HDRFlow | 32.26 | 0.9629 | 0.196 | 76.56 | 116 | 128 |
| DeAltHDR (Default) | **32.55** | **0.9644** | **0.192** | **77.02** | 128 | 152 |
| DeAltHDR-S (compact) | 32.42 | 0.9638 | 0.194 | 76.70 | **102** | **116** |

First, the fundamental mechanism is deblurring via multi-frame fusion. In alternating exposure sequences, blur is non-uniform: a blurred region in a long-exposure frame often corresponds to a sharp region in a complementary short-exposure frame. Our model leverages frame alignment and fusion to effectively remove blur by aggregating sharp information from across the sequence.

Second, the dual-encoder architecture provides exposure-specific feature extraction. The short-exposure encoder learns to extract sharp features while suppressing noise, whereas the long-exposure encoder learns to preserve semantic structure and global context despite blur. This architecture allows the model to take advantage of the distinct strengths of different types of exposure.

Our method demonstrates significantly stronger generalization than traditional, explicitly trained deblurring models. The key distinction lies in its learning objective: instead of learning the blur kernel distribution, our model learns to identify and fuse complementary information across frames. The primary failure mode occurs when all input frames lack sharp reference, which is quite a rare scenario in alternating exposure video.

**Why Does Adaptation Improve Sharpness?** We attribute the sharpness gains to the mitigation of the domain gap between synthetic and real-world blur by our self-supervised adaptation. Specifically, synthetic blur is generated deterministically using linear motion kernels, whereas real-world blur arises from a complex, stochastic interplay of factors like camera shake, object motion, and rolling shutter. Consequently, models trained exclusively on synthetic data are biased and limited to recognizing and recovering simulated blur patterns. Our self-supervised adaptation addresses this gap by minimizing the inconsistency between two reconstructions with differing temporal support: an information-rich, sharper estimate from 5 frames ($\hat{H}_t$) and an information-poor, blurrier one from 3 frames ($\tilde{H}_t$). During adaptation, the model is trained to align the 3-frame reconstruction with the quality of the 5-frame version, thereby learning to recover sharper details from more degraded inputs. Furthermore, we employ motion-augmented sampling (using frames in $t \pm 6$) to avoid over-fitting to synthetic motion patterns. This enhances the model's ability to handle real-world video with a diverse and realistic range of motion blur.

**Why Attention-based Methods can Restore Blur** Although optical-flow-based methods can align blurry frames, their performance is limited under severe or complex motion blur because they rely primarily on local information to estimate dense motion fields. Consequently, they are accurate for smooth motions but are prone to failures when confronted with significant blur and occlusions. In contrast, attention-based methods can leverage global information more fully, which allows them to

Table G: Runtime and FLOP breakdown of DeAltHDR (256×256 patch on RTX A6000).

| Component | Time (ms) | Percentage | FLOPs (G) |
|---|---|---|---|
| Input Processing | 8 | 5.3% | 2.1 |
| Dual Encoders | 42 | 27.6% | 38.4 |
|    Short-exp encoder | 21 | 13.8% | 19.2 |
|    Long-exp encoder | 21 | 13.8% | 19.2 |
| FGMA Alignment | 58 | 38.2% | 52.8 |
|    Flow estimation (SpyNet) | 18 | 11.8% | 8.4 |
|    Mask computation | 4 | 2.6% | 2.2 |
|    Attention (47% pixels) | 36 | 23.7% | 42.2 |
| Frame History Router | 28 | 18.4% | 24.5 |
| Decoder | 12 | 7.9% | 9.8 |
| Output Processing | 4 | 2.6% | 0.4 |
| Total | 152 | 100% | 128 |

Table H: Quantitative comparison of different HDR methods on DeepHDRVideo (Chen et al., 2021) Dataset, Real-HDRV Shu et al. (2024) Dataset and Cinematic Video Froehlich et al. (2014) Dataset.

| Dataset | Methods | PSNR↑ | SSIM↑ | LPIPS↓ | HDR-VDP-2↑ |
|---|---|---|---|---|---|
| DeepHDRVideo | Chen et al. | 42.48 | 0.9620 | 0.184 | 74.80 |
| | LAN-HDR | 41.59 | 0.9472 | 0.181 | 71.34 |
| | NECHDR | 43.44 | 0.9558 | 0.176 | 79.20 |
| | HDRFlow | 43.25 | 0.9520 | 0.174 | 77.29 |
| | DeAltHDR | **43.78** | **0.9572** | **0.172** | **79.32** |
| Real-HDRV | Chen et al. | 36.50 | 0.9262 | 0.192 | 67.56 |
| | LAN-HDR | 38.27 | 0.9334 | 0.184 | 69.24 |
| | NECHDR | 39.23 | 0.9428 | 0.180 | 72.50 |
| | HDRFlow | 38.98 | 0.9434 | 0.179 | 72.32 |
| | DeAltHDR | **40.04** | **0.9489** | **0.175** | **73.14** |
| Cinematic Video | Chen et al. | 35.65 | 0.8949 | 0.172 | 72.09 |
| | LAN-HDR | 38.22 | 0.9100 | 0.162 | 69.15 |
| | NECHDR | 40.59 | 0.9241 | 0.155 | 73.31 |
| | HDRFlow | 39.20 | 0.9154 | 0.158 | 71.05 |
| | DeAltHDR | **40.75** | **0.9245** | **0.152** | **74.22** |

handle low-texture and blur regions more robustly. Consequently, attention mechanisms consistently outperform optical flow in challenging, high-motion scenarios.

Finally, as shown in Table D and Table E, our method outperforms the others whether the reference frame is short-exposure or long-exposure.

# E COMPARISON OF COMPUTATIONAL COSTS

In this section, we provide a detailed analysis of the curve presented in Fig. 1. As shown in Table I, when our method relies solely on optical flow (i.e., $s = 0$), it achieves higher PSNR values while maintaining lower FLOPs and inference time compared to HDRFlow. As the proportion of attention mechanisms increases during inference, both FLOPs and inference time dynamically increase, eventually reaching the configuration where alignment is handled entirely by attention. Compared to other lightweight networks like HDRFlow, our approach achieves higher performance while maintaining competitive computational efficiency.

In addition, we provide a detailed analysis of the relationship between the attention ratio and FLOPs. The relationship between the attention ratio and computational cost (FLOPs) is directly proportional. The total FLOPs cost is a weighted sum of the costs of two operations: efficient flow-based warping

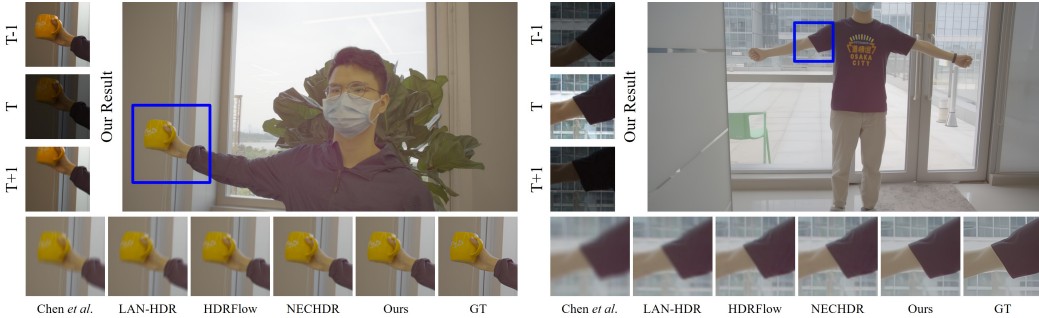

Figure C: Visual comparison on DeepHDRVideo dataset.

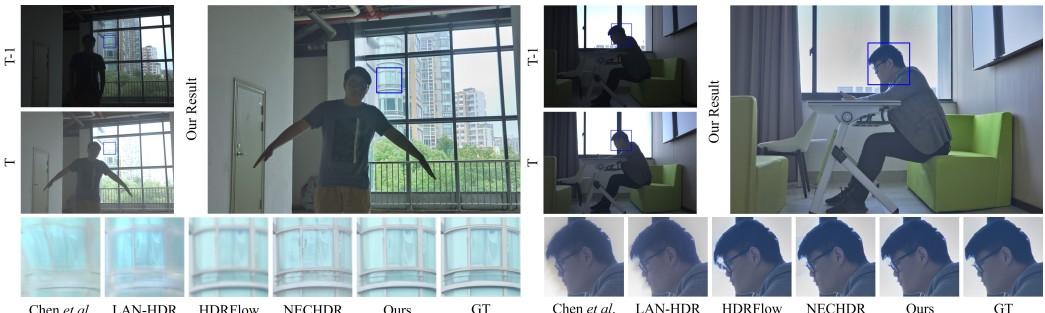

Figure D: Visual comparison on Real-HDRV dataset.

and powerful but expensive attention. Mathematically, it can be derived as: $FLOPs = N \times [(1 - R_{attn}) \times C_{flow} + R_{attn} \times C_{attn}]$, where: $N$ represents the pixel number, $R_{attn}$ is the attention ratio and $C_{flow}$ and $C_{attn}$ are the fixed costs of flow and attention per pixel, with $C_{attn} >> C_{flow}$. As the equation shows, when the attention ratio $R_{attn}$ increases, the overall computational cost increases monotonically, establishing a direct proportional relationship. Each point in the curve in Figure 1 corresponds to a different $R_{attn}$ (controlled by our sensitivity parameter s), and the progression from left to right demonstrates this predictable increase in FLOPs.

Furthermore, we conduct an experiment with a compact model configuration (s=15) that delivers lower inference latency compared to HDRFlow. As shown in Table F, our compact model still achieves better results than HDRFlow, but with fewer FLOPs and shorter inference time. Moreover, we provide more detailed results in Table G. The results show that attention computation constitutes the primary bottleneck, accounting for 38.2% of the total processing time. The dual-encoder architecture also introduces notable overhead, contributing 27.6% to the runtime compared to a single-encoder setup. In comparison, flow estimation is relatively efficient, consuming only 11.8% of the computational budget.

# F EFFECT OF SELF-SUPERVISED ADAPTATION

In this section, we provide more quantitative comparisons in Table J. It can be seen that the proposed adaptation method can bring improvements to both CLIPIQA (Wang et al., 2023) and MANIQA (Yang et al., 2022). Furthermore, only deploying $\mathcal{L}_{ema}$ could prevent parameter updates, while omitting it causes self-supervised training to collapse. This occurs because processing all input frames currently operates without any architectural constraints. Moreover, we evaluate different weighting factors $\lambda_{time}$ for $\mathcal{L}time$. From Tab. J, we can see that when $\lambda_{time}$ is set to 1, the metrics are higher than the other settings. Furthermore, Fig. H shows the visual comparison of adaptation methods. It can be seen that our method achieves more realistic results than BracketIRE.

Table I: Performance comparison between DeAltHDR configurations and HDRFlow.

| Method | $s$ | PSNR↑ | SSIM↑ | LPIPS↓ | FLOPs (G) | Time (ms) |
|--------|-----|-------|-------|--------|-----------|-----------|
| HDRFlow | - | 32.26 | 0.9629 | 0.196 | 116 | 128 |
| DeAltHDR | 0 | 32.42 | 0.9629 | 0.201 | 98 | 94 |
|  | 0.14 | 32.43 | 0.9630 | 0.201 | 101 | 100 |
|  | 0.29 | 32.46 | 0.9633 | 0.198 | 104 | 108 |
|  | 0.43 | 32.48 | 0.9635 | 0.196 | 107 | 119 |
|  | 0.57 | 32.49 | 0.9637 | 0.196 | 111 | 128 |
|  | 0.71 | 32.49 | 0.9640 | 0.196 | 115 | 134 |
|  | 0.86 | 32.52 | 0.9640 | 0.194 | 119 | 138 |
|  | 1 | 32.53 | 0.9642 | 0.192 | 123 | 149 |
|  | 15 | 32.55 | 0.9644 | 0.192 | 128 | 152 |
|  | 30 | 32.57 | 0.9647 | 0.190 | 132 | 162 |
|  | 43 | 32.59 | 0.9649 | 0.188 | 137 | 165 |
|  | 57 | 32.60 | 0.9651 | 0.188 | 142 | 172 |
|  | 71 | 32.60 | 0.9653 | 0.185 | 147 | 186 |
|  | 85 | 32.62 | 0.9657 | 0.184 | 152 | 190 |
|  | 100 | 32.63 | 0.9658 | 0.182 | 159 | 198 |
|  | $\infty$ | 32.65 | 0.9660 | 0.180 | 168 | 224 |

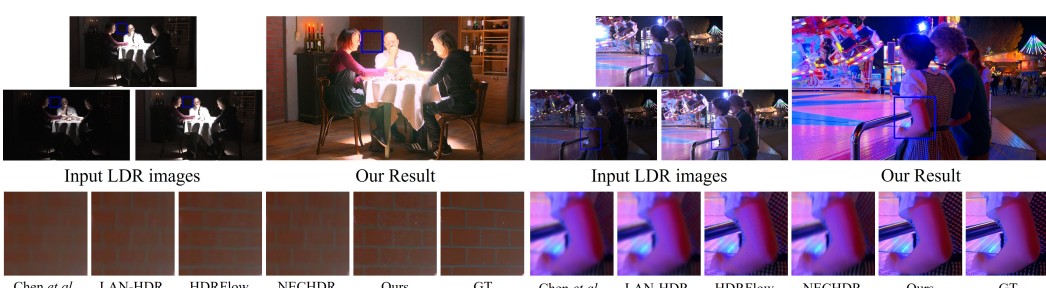

Figure E: Visual comparison on Cinematic Video dataset.

## G    MORE VISUAL RESULTS

We also test our model on the DeepHDRVideo dataset (Chen et al., 2021), Cinematic Video dataset (Froehlich et al., 2014) and the Real-HDRV dataset (Shu et al., 2024). Table H shows the quantitative results on these datasets. It is shown that our method achieves better results than previous HDR video reconstruction methods. Moreover, Fig. C, Fig. D and Fig. E show qualitative results on these datasets. It can be seen that our results have more details and less artifacts. Furthermore, we provide more visual comparisons on both our synthetic dataset and our real-world dataset. Fig. I and Fig. J show the qualitative results. It can be seen that our results have fewer artifacts and more details. Moreover, we perform cross-validation across clean datasets as shown in Table K. The diagonal entries (training set = test set) are the best results and are highlighted in blue; results from our dataset row are second-best and shown in red.

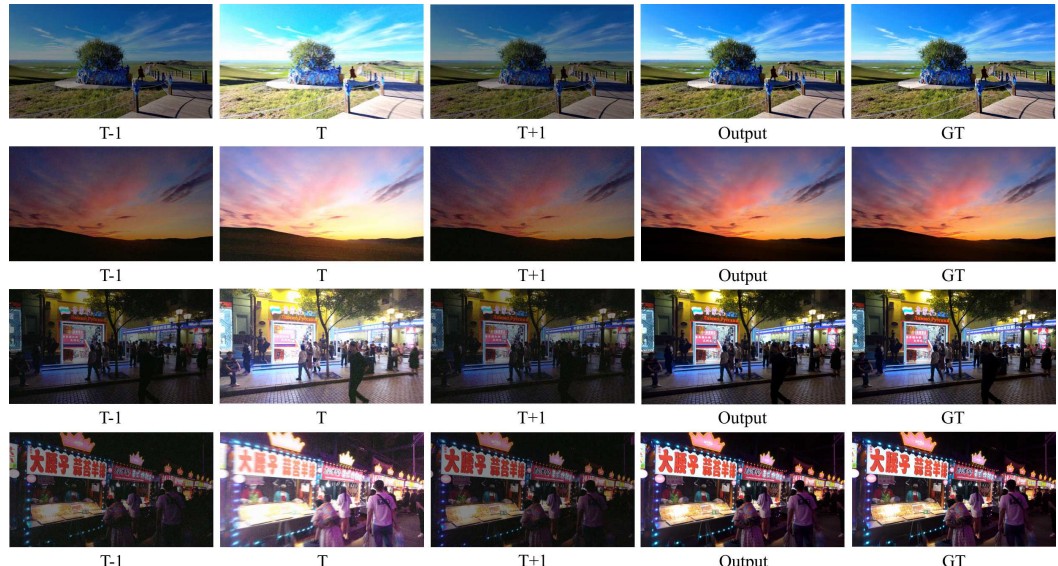

Figure F: Full image results on our synthetic dataset. Our results preserve both the bright areas in short-exposure images and the dark areas in long-exposure images.

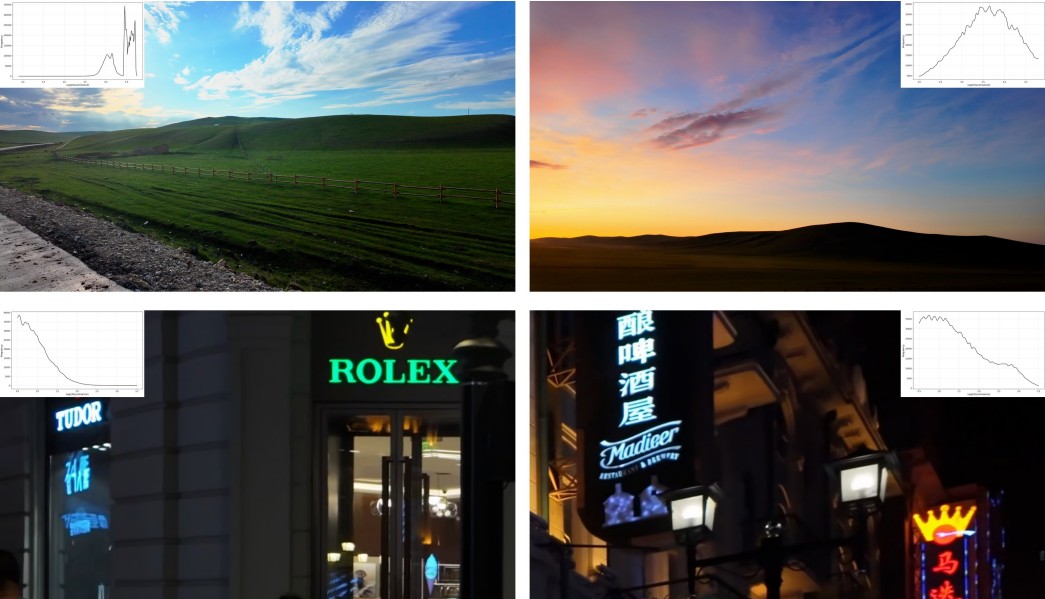

Figure G: Per-scene histograms on our dataset.

Table K: Cross-validation across clean datasets. Diagonal entries (training set = test set) are the best results and are highlighted in blue; results from our dataset row are second-best and shown in red.

| Training Set | Metric | DeepHDR Video | | | Real-HDRV | | | Cinematic Video | | |
| | | NECHDR | HDRFlow | DeAltHDR | NECHDR | HDRFlow | DeAltHDR | NECHDR | HDRFlow | DeAltHDR |
|---|---|---|---|---|---|---|---|---|---|---|
| DeepHDR Video | PSNR | 43.44 | 43.25 | 43.78 | 37.65 | 37.40 | 38.10 | 38.20 | 37.85 | 38.60 |
| | SSIM | 0.9558 | 0.9520 | 0.9572 | 0.9152 | 0.9178 | 0.9284 | 0.9106 | 0.9034 | 0.9216 |
| Real-HDRV | PSNR | 38.05 | 37.80 | 38.50 | 39.23 | 38.98 | 40.04 | 37.50 | 37.10 | 37.95 |
| | SSIM | 0.9184 | 0.9210 | 0.9321 | 0.9428 | 0.9434 | 0.9489 | 0.9066 | 0.9092 | 0.9257 |
| Cinematic Video | PSNR | 38.40 | 37.05 | 38.70 | 37.85 | 37.10 | 38.05 | 40.59 | 40.40 | 40.75 |
| | SSIM | 0.8971 | 0.8890 | 0.9097 | 0.9042 | 0.8872 | 0.9082 | 0.9241 | 0.9154 | 0.9245 |
| Our Dataset | PSNR | 42.64 | 42.32 | 42.78 | 37.82 | 38.42 | 38.79 | 39.98 | 39.80 | 40.11 |
| | SSIM | 0.9532 | 0.9472 | 0.9502 | 0.9048 | 0.9087 | 0.9122 | 0.9373 | 0.9333 | 0.9377 |

Table J: Effect of loss terms for self-supervised real-image adaptation. '−' denotes DeAltHDR trained on synthetic pairs. 'NaN' implies the training collapse.

| $\mathcal{L}_{ema}$ | $\mathcal{L}_{time}$ | CLIPIQA↑/MANIQA↑ |
|---|---|---|
| − | − | 0.2610 / 0.2716 |
| ✓ | ✗ | 0.2610 / 0.2716 |
| ✗ | ✓ | NaN / NaN |
| ✓ | $\lambda_{time} = 0.5$ | 0.2542 / 0.2760 |
| ✓ | $\lambda_{time} = 1$ | **0.2679 / 0.2774** |
| ✓ | $\lambda_{time} = 2$ | 0.2578 / 0.2598 |
| ✓ | $\lambda_{time} = 4$ | 0.2270 / 0.2391 |

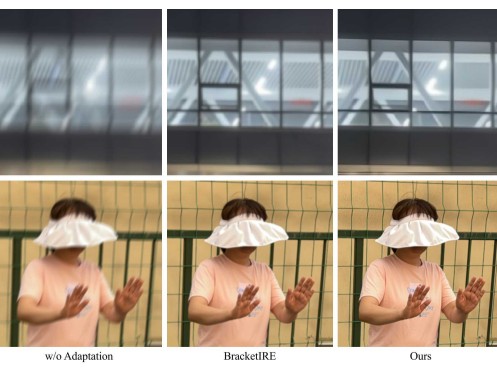

Figure H: Visual comparison of adaptation methods.

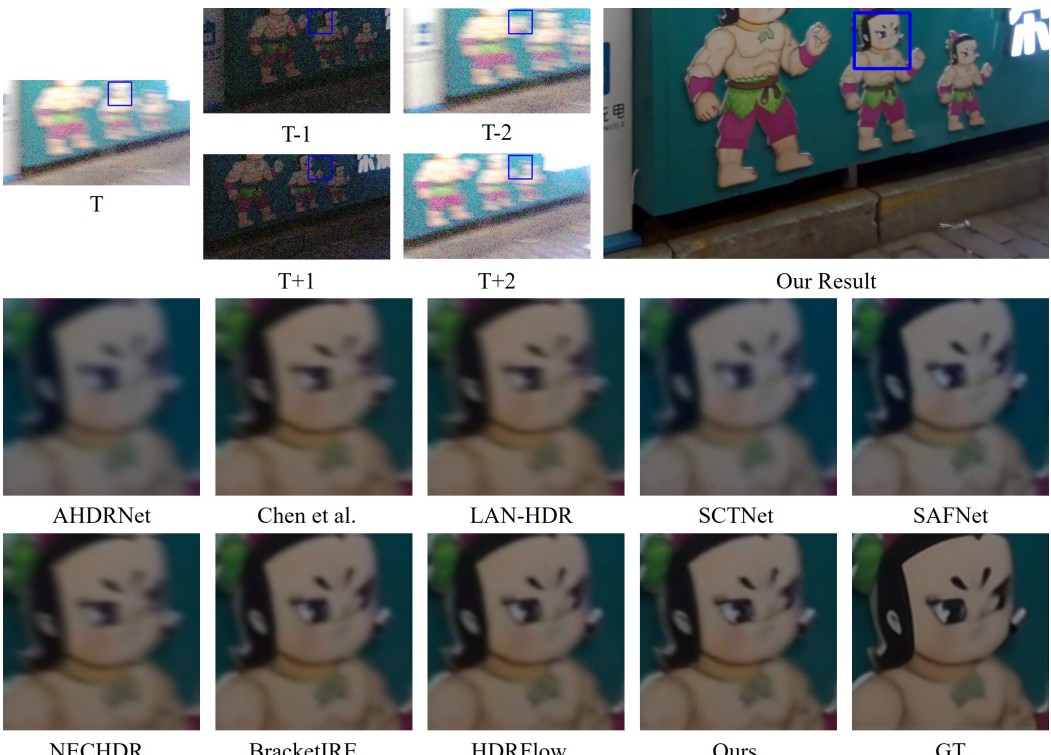

Figure I: Visual comparison on our synthetic dataset.

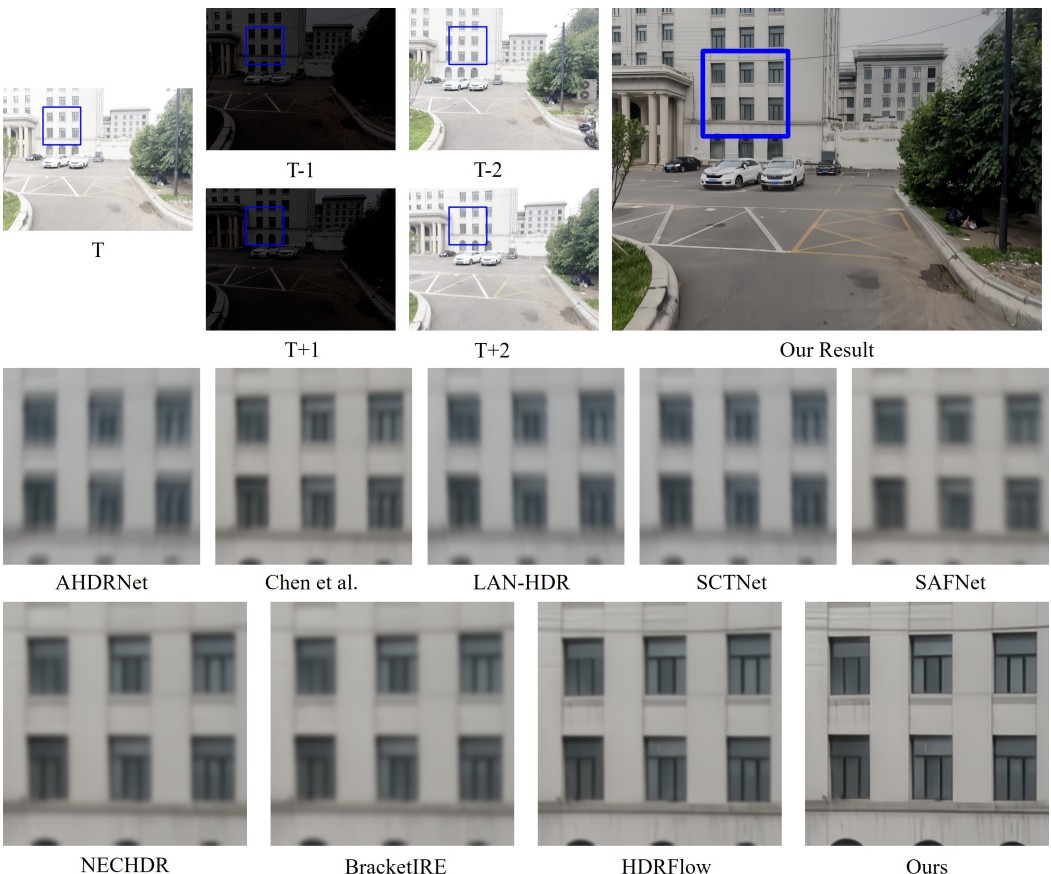

Figure J: Visual Comparison on our real-world dataset.

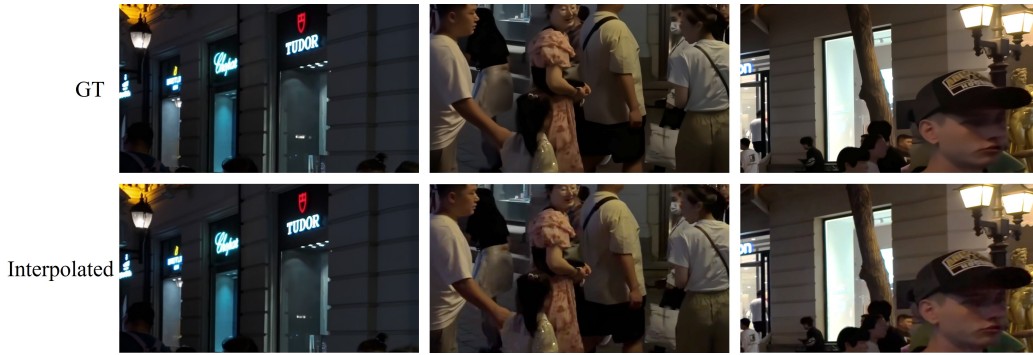

Figure K: Interpolation results on our HDR GT videos.

