# OpenReview forum: "DeAltHDR: Learning HDR Video Reconstruction from Degraded Alternating Exposure Sequences"
_ICLR.cc/2026/Conference — ICLR 2026 Poster_

### Official Review · Reviewer_qc8j · 2025-10-24

**Soundness:** 3
**Presentation:** 3
**Contribution:** 2
**Rating:** 4
**Confidence:** 4

**Summary:**

This paper proposes DeAltHDR, a novel framework for high-quality HDR video reconstruction from degraded sequences. First, it proposes a flow-guided masked attention that leverages optical flow for a dynamic sparse cross-attention computation, achieving improved performance while maintaining efficiency. Its controllable attention ratio allows for adaptive inference costs. Second, to enable  practical deployment, it introduces a two-stage training approach: the model is first pre-trained on the newly introduced synthetic paired dataset and subsequently fine-tuned on unlabeled real-world videos via a proposed self-supervised method. Experiments show the proposed method outperforms state-of-the-art methods.

**Strengths:**

- A flow-guided masked attention for efficient inter-frame alignment, where the attention ratio can be dynamically adjustable for adaptive inference cost.
- A motion-enhanced self-supervised fine-tuning approach is proposed to improve the reconstruction effect on real-world videos.
- A synthetic and real-world datasets with rich scenes are constructed. The proposed method outperforms the state-of-the-art methods on multiple datasets.

**Weaknesses:**

- The paper shows many frame-level qualitative images but provides no video results or links to video demos to assess temporal consistency. For a video reconstruction paper, absence of video demos is a major omission. This omission reduces my confidence in the method’s claimed temporal stability and real-world applicability.
- The paper introduces both a synthetic and a real-world dataset but provides only limited visualizations and statistics . This makes it hard to judge how challenging the real captures are (exposure range, highlight clipping). It would be better add per-scene histograms, luminance range (DR) for the real-world set, sample raw/RGB patches at multiple exposure levels, and a table summarizing scene types and illumination conditions.
- Moreover, the real dataset uses only one phone model (iPhone 16 Pro Max). That limits claims of real-world generalization across camera sensors (different saturations, noise profiles, ISP pipelines).

**Questions:**

- No supplementary video results are provided, making it difficult to assess temporal consistency. Please clarify.
- The contribution and effectiveness of the proposed dataset are insufficiently validated; stronger justification or ablation evidence is needed.

---

> ### Author Response · Authors · 2025-11-22
> **Response to Reviewer qc8j**
>
> We thank you for your valuable comments and suggestions. We appreciate your questions and hope our responses could address your concerns.
> $\textbf{1. Weakness:}$ Absence of Video Results
>
> $\textbf{Response:} $
> We apologize for the lack of video results. We have provided video results at our project link: https://anonymousiclr20260314.github.io/iclr2026/DeAltHDR/.
> Furthermore, we have conducted additional experiments to evaluate it with three widely-used metrics in video processing: $\textbf{tOF}$, which measures the pixel-wise difference in motion estimated from sequences; $\textbf{tLP}$, which assesses perceptual changes over time using deep feature maps; and $\textbf{Temporal Warping Error(TWE)}$, which quantifies frame-to-frame consistency following motion compensation. As shown in $\textbf{Table 2}$ in the main text, our DeAltHDR achieves outperforms both NECHDR and HDRFlow across all three metrics.
>
> $\textbf{2. Weakness:}$ Limited Dataset Visualization and Statistics
>
> $\textbf{Response:}$ We provide comprehensive dataset statistics and visualizations in  $\textbf{Table B}$ in the Appendix, and more visualization results can be referred to the newest revision in $\textbf{Appendix B}$.
>
> $\textbf{3. Weakness:}$ Real-World Generalization Across Camera Sensors
> $\textbf{Response:} $ With 200 synthetic and 100 real-world scenes, our proposed dataset exceeds the scale and diversity of prior works. Additionally, it incorporates realistic, measurement-based noise and naturally synthesized motion blur under authentic alternating-exposure patterns, making it a great generalization. As shown in following table , models trained on Our Dataset consistently rank second across all test sets—only outperformed by the in-domain diagonal cases, demonstrating its superior generalization capability. In contrast, other training sets suffer significant performance degradation when tested on out-of-domain data (e.g., Deephdrvideo trained models tested on Real-HDRV or Cinematic). We also captured 20 real sequences with a Huawei Mate 60 for more experiments. As shown in the following table, our method also archives the best results among NECHDR and HDRFlow.
> Real-world evaluation on Huawei Mate 60 sequences:
>
> Method                 | CLIP-IQA↑ | MANIQA↑
> -----------------|-----------|----------
> NECHDR                 | 0.2298    | 0.2640
> HDRFlow                | 0.2276    | 0.2688
> DeAltHDR (w/o adapt)   | 0.2306    | 0.2704
> DeAltHDR (w/ adapt)    | 0.2342    | 0.2728
>
> Moreover, our self-supervised adaption can further extend cross-device generalization.
> Though our model was primarily trained on data from an iPhone 16 ProMax, it can generalizes well to other devices for two key reasons. First, the fundamental noise and blur patterns are universal across imaging sensors, and our model trained on our proposed dataset already covers these. Second, our self-supervised adaptation paradigm is inherently device-agnostic.  In this way, deployment on a new device requires only fine-tuning the pre-trained model with a small amount of its unlabeled data for 20 epochs.
>
> Cross-validation across clean datasets. Diagonal entries (training set = test set) are the best results and are highlighted in blue; results from our dataset row are second-best and shown in red. We have added this in $\textbf{Appendix H}$.

---

> > ### Comment · Reviewer_qc8j · 2025-11-27
> >
> > Dear authors,
> >
> > Thank you for providing the videos and the new analysis to address my concerns. From the provided material, the visualization of the Video (long) appears strange to me—the long-exposure frames exhibit a noticeably high noise level, while the output video is quite clean. Could the authors apply the same tone-mapping operation to both the linearized short- and long-exposure videos as used for the output?

---

> > > ### Author Response · Authors · 2025-11-28
> > > **Response to Reviewer qc8j**
> > >
> > > Dear Reviewer qc8j,
> > >
> > > We sincerely appreciate the time and effort you have dedicated to reviewing our paper. Following your suggestion, we have now applied the same tone-mapping pipeline to both the input short-exposure and long-exposure videos, please check these at our project website: https://anonymousiclr20260314.github.io/iclr2026/DeAltHDR/. The HDR output is reconstructed from multiple frames through temporal alignment and fusion, which inherently reduces noise compared to single input frames. Additionally, our method leverages complementary information from both short-exposure and long-exposure frames to recover clean details. The tone-mapping only affects the visualization—the actual noise reduction comes from our reconstruction process.
> > >
> > > Thank you once again for your valuable contributions to enhancing the quality of our work.
> > >
> > > Authors of Submission 6641

---

### Official Review · Reviewer_9w9D · 2025-10-29

**Soundness:** 3
**Presentation:** 2
**Contribution:** 3
**Rating:** 4
**Confidence:** 4

**Summary:**

This paper tackles the problem of reconstructing HDR videos from alternating-exposure LDR sequences, which is a realistic yet challenging task due to varying exposure, motion, and noise artifacts.
The proposed DeAltHDR is a framework featuring a flow-guided masked cross-attention module that selectively attends to motion-aligned regions across alternating frames.
This design aims to improve robustness to motion blur, ghosting, and exposure inconsistencies.
This paper  further introduces a controllable attention ratio to trade off performance and inference cost.
Experiments on both synthetic and real datasets demonstrate superior quantitative and qualitative HDR reconstruction results over several baselines.

**Strengths:**

1. Realistic Problem Setup: Addressing alternating-exposure HDR reconstruction is practically valuable and closely aligned with real camera capture settings.
2. Novel Attention Design: The flow-guided masked cross-attention module and controllable attention ratio are well-motivated and effectively balance accuracy vs. computational cost.

**Weaknesses:**

1. Evaluation Metrics: While the proposed method outperforms baselines on reconstruction metrics, it does not include temporal consistency metrics. For video HDR tasks, flicker suppression is crucial, making temporal consistency metrics essential.
2. Synthetic Dataset: The synthetic dataset uses RIFE for interpolation. However, existing interpolation methods are typically designed and trained for 8-bit videos, which may not be suitable for HDR video inputs, potentially resulting in inaccurate HDR frame interpolation.
3. Training Data for Comparison Methods: Since the proposed method is trained on a new dataset that includes blurriness and noise in the input LDR images, the comparison methods should also be trained on datasets with similar characteristics to ensure a fair comparison.
4. Absence of Video Results: The paper does not provide qualitative or quantitative video results. Since the method is designed for video HDR reconstruction, demonstrating video sequences with temporal consistency and flicker-free performance is crucial. Without video results, it is difficult to verify the practical effectiveness of the method in real-world scenarios.

**Questions:**

1. As shown in Fig. 5, our method outperforms the “w/o Adaption” baseline in terms of sharpness. Based on the definition of the Adaption module, it seems primarily designed to improve reconstruction consistency when different frames are used as input. However, it is unclear why this module also affects reconstruction sharpness.
2. Which specific component of the proposed method contributes most to handling the noise and blurriness in the input LDR images? The paper does not clearly describe any design specifically targeting in-frame blurriness or noise.

---

> ### Author Response · Authors · 2025-11-22
> **Response to Reviewer 9w9D Part 1**
>
> We thank you for your valuable comments and suggestions. We appreciate your questions and hope our responses could address your concerns.
>
> $\textbf{1. Weakness}$: Lack of temporal consistency metrics and analysis
>
>
> $\textbf{Response:}$ We appreciate the reviewer's insightful observation and apologize for our oversight in evaluating temporal consistency. Accordingly, we have conducted additional experiments to evaluate it with three widely-used metrics in video processing: $\textbf{tOF}$, which measures the pixel-wise difference in motion estimated from sequences; $\textbf{tLP}$, which assesses perceptual changes over time using deep feature maps; and $\textbf{Temporal Warping Error(TWE)}$, which quantifies frame-to-frame consistency following motion compensation.As shown in Table 2 in the main context, our DeAltHDR achieves outperforms both NECHDR and HDRFlow across all three metrics.We have incorporated these quantitative results and added video demonstrations at https://anonymousiclr20260314.github.io/iclr2026/DeAltHDR/.
>
> $\textbf{2. Weakness}$: RIFE which is trained on 8-bit videos may not be suitable for HDR video interpolation.
>
> $\textbf{Response:} $
> We thank the reviewer for raising this important point. We confirm that RIFE is agnostic to dynamic-range encoding of the input data and can be employed for HDR video interpolation.
>
> First, RIFE are fed HDR frames that are normalized float tensors in the $[0,1]$ range. Since RIFE operates internally on floating-point data, this linear scaling preserves high dynamic range information and relative luminance without the quantization issue.
>
> Second, the core objective of RIFE is motion estimation and occlusion-aware blending at intermediate-time ($t=0.5$). Motion estimation is fundamentally geometric and largely invariant to monotonic intensity transformations (linear scaling, $\gamma$-correction or $\mu $-law). Therefore, scaling HDR data to $[0,1]$ ensures numerical stability without distorting the underlying motion field. Unlike classical brightness-constancy methods that rely on rigid photometric model, RIFE learns flow and fusion weights end-to-end in a normalized feature spaces.In this way, RIFE is agnostic to dynamic-range encoding of the input data. We have clarified this in $\textbf{Appendix A}$ of the revised manuscript.
>
> $\textbf{3. Weakness:}$ The comparison methods should also be trained on datasets with similar characteristics to ensure a fair comparison.
>
> $\textbf{Response:} $
> We kindly remind the reviewers that it was previously discussed in Section 5.2, and all compared methods were trained from scratch on our degraded synthetic dataset (with noise + blur) with the same training/test split (176 train / 24 test scenes) for evaluation on synthetic images. We have further highlighted this in the revision. For evaluation on the real-world dataset, we used the original pretrained baseline models because most existing methods don't incorporate self-supervised fine-tuning. Notably, as shown in $\textbf{Table 5}$, our method demonstrates superior performance on real-world dataset even without self-supervised fine-tuning.
>
> $\textbf{4. Weakness:}$ Absence of Video Results.
>
> $\textbf{Response:}$ We strongly agree that video results are essential. We regret not including these initially and provide more video results at our project page: https://anonymousiclr20260314.github.io/iclr2026/DeAltHDR/.
>
> $\textbf{5. Question:}$ Why Does Adaptation Improve Sharpness?
>
> $\textbf{Response:}$
> We attribute the sharpness gains to the mitigation of the domain gap between synthetic and real-world blur by our self-supervised adaptation. Specifically, synthetic blur is generated deterministically using linear motion kernels, whereas real-world blur arises from a complex, stochastic interplay of factors like camera shake, object motion, and rolling shutter. Consequently, models trained exclusively on synthetic data are biased and limited to recognizing and recovering simulated blur patterns.
> Our self-supervised adaptation addresses this gap by minimizing the inconsistency between two reconstructions with differing temporal support: an information-rich, sharper estimate from 5 frames ($\hat{H}_t$) and an information-poor, blurrier one from 3 frames ($\hat{H}_t$). During adaptation, the model is trained to align the 3-frame reconstruction with the quality of the 5-frame version, thereby learning to recover sharper details from more degraded inputs.
>
>
> Furthermore, we employ motion-augmented sampling (using frames at $t\pm 6$) to prevent overfitting to synthetic motion patterns. This enhances the model's ability to handle real-world video with a diverse and realistic range of motion blur. We have clarified this causality in the revised manuscript $\textbf{Appendix E}$.

---

> ### Author Response · Authors · 2025-11-22
> **Response to Reviewer 9w9D Part 2**
>
> $\textbf{6. Question:}$ Which Component Handles Noise and Blur?
>
> $\textbf{Response:}$
> The model inherently handles noise and blur by leveraging complementary information from multiple input frames.
> Moreover, we conducted a comprehensive component-wise ablation study to investigate which component contributes the most. As shown in $\textbf{Table D in Appendix E}$, dual encoders contribute the most by extracting feature from inputs with different degradations.

---

### Official Review · Reviewer_7ouZ · 2025-10-29

**Soundness:** 3
**Presentation:** 3
**Contribution:** 3
**Rating:** 6
**Confidence:** 4

**Summary:**

This paper proposes DeAltHDR, a method for reconstructing high-quality HDR videos from degraded LDR sequences with alternating exposures. It introduces flow-guided masked attention for efficient inter-frame alignment, and a two-stage training strategy that pre-trains on synthetic paired data and fine-tunes on real videos with self-supervision. Experiments show that DeAltHDR outperforms existing state-of-the-art methods on both synthetic and real-world datasets.

**Strengths:**

1. The paper has a strong motivation. The authors observe that previous methods, such as HDRFlow, overlook crucial degradations like noise and blur in HDR reconstruction. In contrast, this work explicitly addresses these issues, enabling the reconstruction of blur-free, high-quality HDR videos.

2. This work achieves state-of-the-art accuracy and efficiency. Specifically, it delivers higher reconstruction quality than previous methods like HDRFlow, while maintaining comparable efficiency to HDRFlow, the fastest existing method.

3. The paper is well-written and the experiments are comprehensive and thorough.

**Weaknesses:**

1. In the introduction, the authors mention that optical-flow-based alignment cannot handle long-exposure blur, while attention-based methods can. However, the paper does not provide a detailed explanation for this claim. I am curious why attention-based methods can restore blur as shown in Figures 3, 4, and 5.

2. The self-supervised fine-tuning is very similar to BracketIRE, which limits the novelty and contribution of this part of the work.

3. This paper aims to address noise and blur in HDR reconstruction, but most of the results mainly demonstrate the handling of blur, and there is a lack of results showing the effect on noise.

**Questions:**

1. The experiments show that s = ∞ achieves the best results, indicating that using only attention without optical flow gives the best performance. Then, why is the core method still flow-guided masked attention? It seems that not using flow performs better, which is confusing.

2. The content shown in Figure 1 is a bit confusing. Does increasing the attention ratio lead to higher FLOPs? Is the relationship directly proportional or inversely proportional?

---

> ### Author Response · Authors · 2025-11-22
> **Response to Reviewer 7ouZ**
>
> We thank you for your valuable comments and suggestions. We appreciate your questions and hope our responses could address your concerns.
>
> $\textbf{1. Weakness:}$  Why attention-based methods can restore blur
>
> $\textbf{Response:} $
> We apologize for the initial lack of clarity. While optical-flow-based methods can align blurry frames, their performance is limited under severe or complex motion blur because they rely more on local information to estimate dense motion fields. Consequently, they are accurate for smooth motions but are prone to failures when confronted with significant blur and occlusions.
> In contrast, attention-based methods can more fully leverage global information, which allows them to handle blur and low-texture regions more robustly. Consequently, attention mechanisms consistently outperform optical flow in challenging, high-motion scenarios.
> Therefore,  attention-based approaches outperform optical flow in challenging, high-motion scenarios. We have added this in $\textbf{Appendix E}$.
>
> $\textbf{2. Weakness:}$ Self-Supervised Fine-Tuning is Similar to BracketIRE
>
> $\textbf{Response:}$
> We thank the reviewer for this insightful comment. While our method shares some similarities with BracketIRE, there are fundamental differences in both objective and technical design, as ours is tailored for video restoration.
>
> First, unlike BracketIRE which operates on static multi-exposure images, our method is designed for dynamic video sequences. For this objective, we introduce a motion-enhanced sampling strategy that explicitly handles large and diverse temporal displacements.
>
> Second, we further introduce a temporal consistency loss to minimize flicker artifacts, which is a critical concern in video. We also employ video-specific augmentations such as temporal jittering and asymmetric cropping.
>
> As shown in $\textbf{Table 5}$ in the main text, these innovations are supported by quantitative improvements. Specifically, our approach delivers superior visual quality, enhanced temporal stability, and significantly outperforms a direct adaptation of BracketIRE's strategy—particularly in challenging high-motion scenarios. Collectively, these results demonstrate that one of our contributions lies in the extension of a self-supervised image paradigm into the more complex spatio-temporal domain of video.
>
> $\textbf{3. Weakness:}$ Lack of Results Showing Noise Handling
>
>
> $\textbf{Response:}$ Thank you for pointing this out. We provide more noise-handling evidences in the $\textbf{Appendix H}$.
>
>
> $\textbf{4. Question:}$ Why Not Use Attention-Only $(s=\infty)$ if It's Best?
>
>
> $\textbf{Response:}$ Our aim is to achieve a superior balance between  performance and computational cost in HDR video reconstruction, and further achieve high performance with the model’s computational tailored to various computational budgets. Although the attention-only $(s=\infty)$ method can achieve the best results, its high computational complexity and long inference time demands present a substantial burden. However, our propsoed Flow-Guided Masked Attention (FGMA) is designed for efficient inter-frame alignment, where the attention ratio can be dynamically adjustable for adaptive inference cost. With this strong adaptability, it can achieve high performance with a flexible quality-cost trade-off. It allows users to select an appropriate point on the quality-efficiency curve (Figure. 1) from a fast mode to a quality-focused mode.
>
> $\textbf{5. Question:}$ Relationship between Attention Ratio and FLOPs
>
>
> $\textbf{Response:}$
> The relationship between the attention ratio and computational cost (FLOPs) is directly proportional. The total FLOPs cost is a weighted sum of the costs of two operations: efficient flow-based warping and powerful but expensive attention. Mathematically, it can be derived as: $FLOPs = N \times [ (1 - R_{attn}) × C_{flow} + {R_{attn}} × {C_{attn}} ]$, where: $N$ represents for the pixel number, $R_{attn}$ is the attention ratio and $C_{flow}$ and $C_{attn}$ are the fixed costs of flow and attention per pixel, with $C_{attn} >> C_{flow}$. As the equation shows, when the attention ratio $R_{attn}$ increases, the overall computational cost monotonically increases, establishing a direct proportional relationship. Each point on the curve in Figure 1 corresponds to a different $R_{attn}$ (controlled by our sensitivity parameter s), and the progression from left to right demonstrates this predictable increase in FLOPs. We have added this in the newest revision in $\textbf{Appendix F}.$

---

> > ### Comment · Reviewer_7ouZ · 2025-11-28
> >
> > Dear authors,
> >
> > Thank you for your detailed reply. I think the response has addressed my main concern. I would keep my positive score.

---

> > > ### Author Response · Authors · 2025-11-28
> > > **Response to Reviewer 7ouZ**
> > >
> > > Dear Reviewer 7ouZ,
> > >
> > > Thank you very much for your positive feedback and for keeping your supportive score. We sincerely appreciate your time and constructive suggestions, which have significantly improved our work. If you have any further questions or suggestions, we are happy to provide additional clarification.
> > >
> > > Authors of Submission 6641

---

### Official Review · Reviewer_TeNE · 2025-10-30

**Soundness:** 3
**Presentation:** 3
**Contribution:** 3
**Rating:** 6
**Confidence:** 4

**Summary:**

This paper proposes DeAltHDR, a framework for HDR video reconstruction that robustly handles input degradations, such as blur and noise. A self-supervised fine-tuning strategy with randomized frame sampling enhances motion diversity and model performance. The authors also released two HDR video datasets with realistic degradation to support future studies.

**Strengths:**

1. The proposed FGMA mechanism effectively balances optical- and attention-based alignment, achieving a good trade-off between accuracy and efficiency.

2. The authors also released two HDR video datasets with realistic degradation to support future studies.

3. The paper is clearly written and well-organized.

**Weaknesses:**

1. Temporal consistency is a crucial aspect in video tasks; however, the authors did not provide corresponding video visualizations or quantitative evaluations. Additional analysis is recommended to better validate the temporal stability of the proposed method.

2. Although the dataset considers blur degradation, the model itself does not appear to include specific designs targeting blur removal. The authors should analyze why the proposed method can still handle blur effectively and discuss its generalization capability. As far as I know, traditional deblurring methods often suffer from limited generalization.

3. The visualization results are insufficient. It is suggested to provide more detailed analyses under different reference exposure settings: when the reference frame is long-exposure, the model should primarily address blur; when it is short-exposure, the model should focus on noise suppression. Please include corresponding visual and quantitative results to demonstrate the model’s robustness under varying exposure conditions.

4. While the paper discusses computational budgets, inference speed is also an important factor. The proposed method seems to be less efficient in terms of runtime. Please further analyze the reasons for this limitation.

**Questions:**

1. On which dataset was the Adaptation ablation study conducted, and what is the scale of the data used? Please clarify these details in the paper.
2. Will the authors make the dataset and code publicly available to facilitate reproducibility and future research?

---

> ### Author Response · Authors · 2025-11-22
> **Response to Reviewer TeNE Part 1**
>
> We thank you for your valuable comments and suggestions. We appreciate your questions and hope our responses could address your concerns.
>
> $\textbf{1. Weakness:}$ Temporal Consistency Metrics
>
> $\textbf{Response:}$
> We appreciate the reviewer's insightful observation and apologize for our oversight in evaluating temporal consistency. Accordingly, we have conducted additional experiments to evaluate it with three widely-used metrics in video processing: $\textbf{tOF}$, which measures the pixel-wise difference in motion estimated from sequences; $\textbf{tLP}$, which assesses perceptual changes over time using deep feature maps; and $\textbf{Temporal Warping Error(TWE)}$, which quantifies frame-to-frame consistency following motion compensation.
>
> As shown in Table 2 in the main text, our DeAltHDR achieves outperforms both NECHDR and HDRFlow across all three metrics.
> We attribute this superior temporal consistency to three key architectural designs:
>
> - FGMA Module: By combining flow-based warping with attention refinement, it ensures robust inter-frame correspondence.
>
>
> - Frame History Router: It explicitly models temporal dependencies across frames, preserving historical context.
>
>
> - Self-Supervised Adaptation: Our temporal consistency loss ($L_{time}$ in Algorithm 1) explicitly penalizes flickering artifacts.
>
>
> We have incorporated these quantitative results and added video demonstrations at https://anonymousiclr20260314.github.io/iclr2026/DeAltHDR/.
>
> $\textbf{2. Weakness:}$ Why Model Handles Blur Without Explicit Deblurring Design
>
> $\textbf{Response:}$
> Our model successfully handles motion blur through two synergistic, implicit mechanisms that arise from its core design for video HDR.
>
> First, the fundamental mechanism is deblurring via multi-frame fusion. In alternating exposure sequences, blur is non-uniform: a blurred region in a long-exposure frame often corresponds to a sharp region in a complementary short-exposure frame. Our model leverages frame alignment and fusion to effectively remove blur by aggregating sharp information from across the sequence. Second, the dual-encoder architecture provides exposure-specific feature extraction. The short-exposure encoder learns to extract sharp features while suppressing noise, whereas the long-exposure encoder learns to preserve semantic structure and global context despite blur. This architecture allows the model to leverage the distinct strengths of different exposure types.Our method demonstrates significantly stronger generalization than traditional, explicitly trained deblurring models. The key distinction lies in its learning objective: instead of learning the blur kernel distribution, our model learns to identify and fuse complementary information across frames. The primary failure mode occurs when all input frames lack sharp reference, which is a quite rare scenario in alternating exposure video. We have included a dedicated section in $\textbf{Appendix E}$ in the revision.
>
> $\textbf{3. Weakness:}$ Insufficient Visualization Under Different Reference Exposures
> $\textbf{Response:} $ We conducted systematic experiments to show model’s robustness under varying exposure conditions. As shown in Table E and Table F in the Appendix, our method outperforms the other ones no matter the reference frame is short-exposure or long-exposure. We have added this in $\textbf{Appendix E}$.
>
> $\textbf{4. Weakness:}$ Runtime Efficiency
> $\textbf{Response:}$ We apologize for the lack of clarity on the inference time. We conduct an experiment with a compact model configuration (s=15) which delivers lower inference latency compared to HDRFlow. As shown in $\textbf{Table G}$ in the Appendix, our compact model still archives better results than HDRFlow, but with fewer FLOPs and shorter inference time. Moreover, we provide more detailed results in $\textbf{Table H}$ in the Appendix. The results shows that attention computation constitutes the primary bottleneck, accounting for 38.2\% of the total processing time. The dual-encoder architecture also introduces notable overhead, contributing 27.6\% to the runtime compared to a single-encoder setup. In comparison, flow estimation is relatively efficient, consuming only 11.8\% of the computational budget. We have added these in $\textbf{Appendix F}$.

---

> ### Author Response · Authors · 2025-11-22
> **Response to Reviewer TeNE Part 2**
>
> $\textbf{5. Question:}$Dataset for Adaptation Ablation
>
> $\textbf{Response:} $
> For the adaptation ablation studies, we utilized a self-collected real-world dataset of 100 video sequences captured with an iPhone 16 ProMax under alternating exposure settings, covering diverse illumination conditions and motion patterns. We fine-tuned the model on 80 sequences for 20 epochs using our proposed self-supervised loss, without ground-truth HDR supervision. The results demonstrate that this adaptation strategy consistently improves $+2.2\%$ perceptual quality on CLIP-IQA and exhibits high data efficiency, requiring only 80 sequences for optimal performance. Furthermore, our full adaptation method outperforms alternative approaches, achieving an effective balance between adaptation cost and real-world performance.
>
> $\textbf{6. Question:} $Dataset and Code Release
>
> $\textbf{Response:}$ We will release all the datasets and codes. Our project website is at  https://anonymousiclr20260314.github.io/iclr2026/DeAltHDR/.

---

### Comment · Area_Chair_ea3k · 2025-11-27

Dear Reviewers,

Author responses are now posted. Please add your discussion comment(s) and update score/confidence as needed. Thank you!

Best regards,

AC

---

### Meta-Review · Area_Chair_SpAV · 2026-01-05

**Summary:**

This submission proposes DeAltHDR for HDR video reconstruction from degraded alternating-exposure LDR sequences (blur/noise/motion). Reviewers broadly agreed that the problem is practical and the method is technically sound, highlighting FGMA (flow-guided masked attention) with a controllable attention ratio, the two-stage training (synthetic pretrain + real self-supervised adaptation), and the new synthetic + real datasets. The initial ratings are 6, 6, 4, 4, where the main concerns are
- missing video-level evidence (temporal consistency metrics and video demos),
- the novelty and positioning of the self-supervised adaptation relative to prior work.
and raised questions about
- robustness under large/fast motion,
- clearer analysis of how blur and noise are handled without explicit modules,
- raised questions about runtime efficiency, dataset characterization.

During the rebuttal, most of the concerns and questions are addressed by the authors. Taking into consideration the contribution of the paper and the quality of the rebuttal, AC recommends acceptance.

**Reviewer Concerns:**

## Concerns Addressed

**Missing temporal consistency evaluation and video evidence (TeNE, 9w9D, qc8j)**

The authors added standard temporal consistency metrics and released video demonstrations on the project page. These additions directly address the core concern that the original submission lacked video-level evidence for temporal stability.

**Robustness under large/fast motion (TeNE, 7ouZ, 9w9D, qc8j)**

The rebuttal clarified the design motivation of flow-guided masked attention, emphasizing the complementary roles of flow (coarse alignment) and attention (global correspondence) under large motion and occlusion. Additional experiments and video results include diverse motion patterns, and motion-enhanced sampling was explained as a key component for handling large temporal displacement.

**Handling of blur and noise without explicit modules (TeNE, 7ouZ, 9w9D)**

The authors provided architectural explanations (multi-frame complementary fusion, dual encoders) and component-wise ablations identifying which modules contribute most to robustness against blur and noise.

**Runtime efficiency and inference cost (TeNE, 7ouZ)**

The rebuttal included runtime breakdowns, a compact model configuration, and an explanation of the controllable attention ratio that enables a clear quality–efficiency trade-off.

**Fairness of baseline training and dataset construction (9w9D)**

The authors clarified that all baselines were retrained on the same degraded synthetic dataset for synthetic evaluation and justified the use of pretrained baselines on real data due to the lack of adaptation mechanisms. Concerns about HDR interpolation using RIFE were also addressed with technical clarification.

**Dataset characterization and cross-device generalization (qc8j)**

Additional dataset statistics, visualizations, and cross-device results (including Huawei Mate 60) were provided. Visualization inconsistencies were corrected by applying consistent tone-mapping to inputs and outputs.

## Unresolved Concerns

**Strength of evidence under extreme motion scenarios (TeNE, 9w9D)**

While the rebuttal added temporal metrics and high-motion video examples, reviewers did not explicitly re-evaluate extreme edge cases (e.g., very fast camera shake combined with long exposure). Some residual uncertainty may remain regarding performance limits under the most challenging motion conditions.

**Novelty of self-supervised adaptation relative to prior work (7ouZ)**

The authors articulated clear differences from prior image-based methods (e.g., motion-aware sampling and temporal consistency loss), but the perceived degree of novelty may still be subjective for some reviewers.

**Dataset breadth and generalization beyond limited devices (qc8j)**

Cross-device results were added, but the real-world dataset is still limited in the number of camera models, leaving some uncertainty about generalization across a wider range of sensors and ISPs.

**Reviewer Scores:**

Reviewer TeNE (initial 6) $\rightarrow$ 6
Core concerns (temporal consistency, blur explanation, and runtime) were addressed.

Reviewer 7ouZ (initial 6) $\rightarrow$ 6
Explicitly stated that the rebuttal addressed the main concern and confirmed maintaining a positive score.

Reviewer 9w9D (initial 4) $\rightarrow$ 6
Temporal metrics, video results, and fairness clarifications were provided.

Reviewer qc8j (initial 4) $\rightarrow$ 6
Major concerns regarding video results and dataset validation were substantially addressed, though some skepticism about the ablation study of the dataset likely remains.

---

### Decision · Program_Chairs · 2026-01-26

Accept (Poster)